# DYNAMIC INSTANCE HARDNESS

## ABSTRACT

We introduce *dynamic instance hardness* (DIH) to facilitate the training of machine learning models. DIH is a property of each training sample and is computed as the running mean of the sample's instantaneous hardness as measured over the training history. We use DIH to evaluate how well a model retains knowledge about each training sample over time. We find that for deep neural nets (DNNs), the DIH of a sample in relatively early training stages reflects its DIH in later stages and as a result, DIH can be effectively used to reduce the set of training samples in future epochs. Specifically, during each epoch, only samples with high DIH are trained (since they are historically hard) while samples with low DIH can be safely ignored. DIH is updated each epoch only for the selected samples, so it does not require additional computation. Hence, using DIH during training leads to an appreciable speedup. Also, since the model is focused on the historically more challenging samples, resultant models are more accurate. The above, when formulated as an algorithm, can be seen as a form of curriculum learning, so we call our framework DIH curriculum learning (or DIHCL). The advantages of DIHCL, compared to other curriculum learning approaches, are: (1) DIHCL does not require additional inference steps over the data not selected by DIHCL in each epoch, (2) the dynamic instance hardness, compared to static instance hardness (e.g., instantaneous loss), is more stable as it integrates information over the entire training history up to the present time. Making certain mathematical assumptions, we formulate the problem of DIHCL as finding a curriculum that maximizes an unknown multi-set function $f(\cdot)$ from its partial observations, and derive an approximation bound for a DIH-produced curriculum relative to the optimal curriculum. Empirically, DIHCL-trained DNNs significantly outperform random mini-batch SGD and other recently developed curriculum learning methods in terms of efficiency, early-stage convergence, and final performance, and this is shown in training several state-of-the-art DNNs on 11 modern datasets.

## 1 INTRODUCTION

We study the dynamics of training a machine learning model, and in particular, the difficulty a model has over time (i.e., training epochs) in learning each sample from a training set. To this end, we introduce the concept of "dynamic instance hardness" (DIH) and propose several metrics to measure DIH, all of which share the same form as a running mean over different instantaneous sample hardness measures. Let $a_t(i)$ be a measure of instantaneous (i.e., at time $t$) hardness of a sample, where $i$ is a sample index and $t$ is a time iteration index (typically a count of mini-batches that have been processed). In previous work, $a_t(i)$ has been called the "instance hardness" (Smith et al., 2014) corresponding to $1 - p_w(y_i|x_i)$, i.e., the complement of the posterior probability of label $y_i$ given input $x_i$ for the $i^{\text{th}}$ sample under model $w$. We introduce three different notions of instantaneous instance hardness in this work: (A) the loss $\ell(y_i, F(x_i; w_t))$, where $\ell(\cdot, \cdot)$ is the loss function and $F(\cdot; w)$ is the model with parameters $w$, (B) the loss change $|\ell(y_i, F(x_i; w_t)) - \ell(y_i, F(x_i; w_{t-1}))|$ between two consecutive time steps, and (C) the prediction flip $\mathbb{1}[\hat{y}_i^t = \hat{y}_i^{t-1}]$, where $\hat{y}_i^t$ is the prediction of sample $i$ in step $t$, e.g., $\operatorname{argmax}_j F(x_i; w_t)[j]$ for classification. Our (A) corresponds closely to the "instance hardness" of Smith et al. (2014). However, our (B) and (C) require information from previous time steps. Nevertheless, we consider (A), (B), and (C) all variations of instantaneous instance hardness since they use information from only a very local time window around training iteration $t$. Dynamics is achieved when we compute a running average over instantaneous instance

hardness, computed recursively as follows:

$$r_{t+1}(i) = \begin{cases} \gamma \times a_t(i) + (1 - \gamma) \times r_t(i) & \text{if } i \in S_t \\ r_t(i) & \text{else}, \end{cases} \qquad (1)$$

where $\gamma \in [0, 1]$ is a discount factor, $S_t \subseteq V$, and $V = [n]$ is the set of all $n$ training sample indices. $S_t$ is the set of sample selected for training at time $t$ by some method (e.g., a DIH-based curriculum learning (DIHCL) algorithm we introduce and study below) or simply a random batch. In general, $S_t$ should be large early during training, but as $r_t(i)$ decreases to small values for many samples, choosing significantly smaller $S_t$ is possible to result in faster training and more accurate models.

We find that $r_t(i)$ can vary dramatically between different samples since very early stage (with small $t$). One can think of this as some samples being more memorable and are retained more easily, while other samples are harder to learn and retain. In addition, the predictions of the hard samples are less stable under changes in optimization parameters (such as the learning rate).

More importantly, once a sample's $r_t(i)$ is established (i.e., once $t$ is sufficiently but not unreasonably large) each sample tends to maintain its DIH properties. That is, a sample's DIH value converges relatively quickly to its final relative position amongst all of the samples DIH values. For example, if a sample's DIH becomes small (i.e., meaning the sample is easily learned), it stays small relative to the other samples, or if it becomes large DIH (i.e., the sample is difficult to learn), it stays there. I.e., once $r_t(i)$ for a sample has converged, its DIH status is retained throughout the remainder training. We can therefore accurately identify categories of sample hardness relatively early in the course of training. This suggests a natural curriculum learning strategy where $S_t$ corresponds mostly to those samples that are hard according to $r_{t-1}(i)$. In other words, the model concentrates on that which it finds difficult. This is similar to strategies that improve human learning, such as the Leitner system for spaced repetition (Leitner, 1970). This is also analogous to boosting (Schapire, 1990) — in boosting, however, we average the instantaneous sample performance of multiple weak learners at the current time, while in DIHCL we average the instantaneous sample performance of one strong learner over the training history.

As mentioned above, instance hardness has been studied before (Smith et al., 2014; Prudencio et al., 2015; Smith & Martinez, 2016) where it corresponds to the complement posterior probability. More recently, instance hardness has also been studied as an average over training steps in Toneva et al. (2019) where the mean of prediction flips over the entire training history is computed. We note that Toneva et al. (2019) is a special case of DIH in Eq. (1) with $\gamma = 1/t+1$ and $t = T$, where $T$ is the total number of training steps. Our study generalizes Toneva et al. (2019) to the running dynamics computed during training. This therefore leads to a novel curriculum learning strategy and also steps towards a better theoretical understanding of curriculum learning. Also, in Toneva et al. (2019), a small neural net is trained beforehand to determine the hard samples, and this is then used to train large neural nets. In our approach, we take the average over time of $a_t(i)$, which requires no additional model or inference steps and hence is computationally trivial.

Another observation we find is that $r_t(i)$, for any sample, tends to monotonically decrease with $t$ for any $i$. This means, not surprisingly, that during training samples become easier in terms of small DIH (i.e., they are better learned). This also means that easy samples stay easy throughout training, and hard samples also become easier the more we train on them. If we also make (admittedly) a mathematical leap, and assume that $r_t(i)$ is generated by the marginal gain of an unknown diminishing returns function $f(\cdot)$ that measures the quality of any curriculum, we can formulate DIHCL as an online learning problem that maximizes the unknown $f(\cdot)$ by observing its partial observation $r_t(i)$ over time for each $i$. Here, $f$ is defined over an integer lattice and has a diminishing returns property, although the function is accessible only via the gains of every element. This formulation provides a setting where the quality of the learnt curriculum is provably approximately good.

As will be shown below, DIHCL performs optimization in a greedy manner. At each time step $t$, DIHCL selects a subset $S_t$ of samples using $r_t(i)$ where the hard samples have higher probabilities of being selected relative to the easy samples. The model is then updated based only on the selected subset $S_t$ rather than $V$, which requires performing inference (e.g., a forward pass of a DNN) only on $S_t$. This therefore leads to a speedup to the extent that $|S_t| \ll |V|$. The inference produces new instantaneous instance hardness $a_t(i)$ that is then used to update $r_{t+1}(i)$ as in Equation 1. To encourage exploration, improve stability, and get an initial estimate of $r_t(i)$ for all $i \in V$, during the first few epochs, DIHCL sweeps through the entire training set. We provide several options for DIH-weighted subset sampling, which introduces different types of randomness in the selection since

randomness is essential in optimizing non-convex problems. Under certain additional mathematical assumptions, we also give theoretical bounds on the curriculum achieved by DIHCL compared to the optimal curriculum. We empirically evaluate several variants of DIHCL and compare them against random mini-batch SGD as well as against recent curriculum learning algorithms, and test on 11 datasets including CIFAR10, CIFAR100, STL10, SVHN, Fashion-MNIST, Kuzushiji-MNIST, Food-101, Birdsnap, FGVC Aircraft, Stanford Cars and ImageNet. DIHCL shows an advantage over other baselines in terms both of time/sample efficiency and test set accuracy.

## 1.1 RELATED WORK

Early curriculum learning (CL) (Khan et al., 2011; Basu & Christensen, 2013; Spitkovsky et al., 2009) work shows that feeding an optimized sequence of training sets (i.e., a curriculum), that can be designed by a human expert (Bengio et al., 2009), into the training algorithms can improve the models' performance. Self-paced learning (SPL) (Kumar et al., 2010; Tang et al., 2012a; Supancic III & Ramanan, 2013; Tang et al., 2012b) chooses the curriculum adaptive to some instance hardness (e.g., per-sample loss) during training. SPL selects samples with smaller losses, and gradually increases the subset size over time to cover all the training data. Self-paced curriculum learning (Jiang et al., 2015) combines the human expert in CL and loss-adaptation in SPL. SPL with diversity (SPLD) (Jiang et al., 2014) adds a negative group sparse regularization term to SPL and increases its weight to increase selection diversity. Machine teaching (Khan et al., 2011; Zhu, 2015; Patil et al., 2014) aims to find the optimal and smallest training subset leading to similar performance as training on all the data. Minimax curriculum learning (MCL) (Zhou & Bilmes, 2018) argues that the diversity of samples is more critical in early learning since it encourages exploration, while difficulty becomes more useful later. It also uses a form of instantaneous instance hardness (the loss) but is not dynamic like DIH, and formulates optimization as a minimax problem. Compared to the above methods, DIHCL has the following advantages: (1) DIHCL improves the efficiency of CL since extra inference on the entire training set per step is not required; and (2) DIHCL uses DIH as the metric for hardness which is a more stable measure than instantaneous hardness.

Our paper is also related to Zhang et al. (2017), which refers to overfitting in noisy data. Our observations suggest that the learning of simple patterns (Arpit et al., 2017) happen mainly amongst the easy memorable early during in training (additional discussion is given in the Appendix and Figure 5). Our paper is also distinct from catastrophic forgetting (Kirkpatrick et al., 2017), which considers sequential learning of multiple tasks, where later learned tasks make the model forget what has been learned from earlier tasks. In our work, we consider single task learning and show that easy samples remain easy.

If we make certain additional mathematical assumptions (as we do in our theoretical discussion below), our work is related to online submodular function optimization. Specific forms have been studied including maximization (Streeter & Golovin, 2009; Chen & Krause, 2013), maximization in the bandit setting with noisy feedback (Chen et al., 2017), and continuous submodular function maximization (Chen et al., 2018b;a).

The work most related to ours, perhaps, is the work on instance hardness (Smith et al., 2014; Prudencio et al., 2015; Smith & Martinez, 2016), where the hardness of a sample corresponds to the complement posterior probability, as discussed above. Also, a special case of DIH was studied in Toneva et al. (2019): they compute DIH after training completes, and show that removing the easy samples (those having the smallest DIH over training set) leads to less degradation on generalization performance than removing random samples. By contrast, our study of DIH focuses on its dynamics during training.

## 2 DYNAMIC INSTANCE HARDNESS

We start out by conducting an empirical study of DIH in DNN training. We train a WideResNet with depth of 28 and width factor 10 on CIFAR10 dataset, and apply a modified cosine annealing learning rate schedule (Loshchilov & Hutter, 2017) for multiple episodes of increasing length (300 epochs in total) and target learning rate decaying. We contend that a cyclic learning rate suits our study because: (1) it includes the most commonly used monotone decreasing schedule since the learning rate in each cycle is decreasing; (2) compared to monotone decreasing schedule, it can uncover the properties of DIH in more scenarios such as increasing learning rate and different decaying speeds of the learning rate. In the study, we test two type of instantaneous instance hardness, where $a_t(i)$ is either prediction flips or loss (i.e., cases (A) and (C) in the previous section). We visualize $r_t(i)$ for

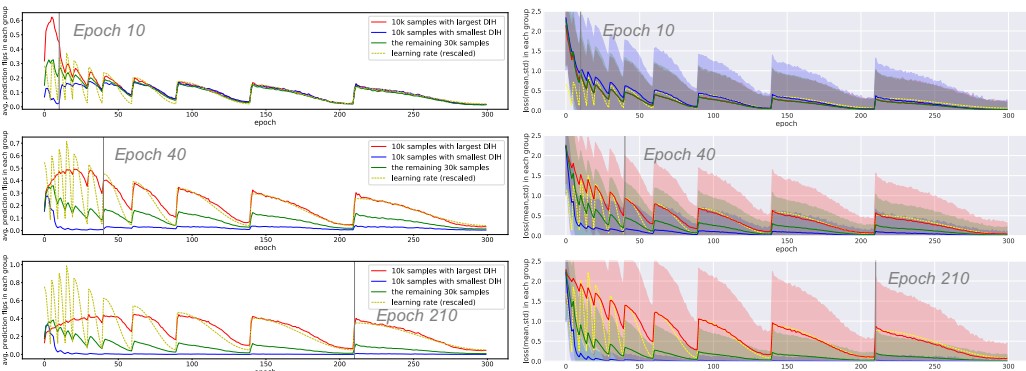

Figure 1: **LEFT:** Averaged prediction-flip and **RIGHT:** losses (mean and std.) of the three groups of samples partitioned by a DIH metric (i.e., running mean of prediction-flip) computed at epoch 10,40 and 210 during training WideResNet-28-10 on CIFAR10. DIH in early stage (Epoch 40) can predict the forgettable and memorable samples for later stages. The failed partition based on Epoch 10 DIH implies the importance of sufficient exploration to accurately measure hardness over time.

all $i \in [50000]$ training samples, but divide $[50000]$ them into three groups according to $r_t(i)$ (with $a_t(i)$ as prediction flips), and we do this at epoch either 10, 40, or 210. For example, at epoch 40, the 10,000 samples with the largest $r_{40}(i)$ comprise the first group, the 10,000 samples ones with the smallest $r_{40}(i)$ comprise the next group, and the remaining 30,000 samples comprise the final group. In Figure 1, we plot the dynamics for the average prediction flips over each group (left plot) and the mean/standard deviation of loss in each group (right plot).

We observe that samples with small $r_t(i)$ are learned quickly in the early epochs and their losses remain small and predictions almost unchanged thereafter, indicating they are easy over time. By contrast, the samples with large $r_t(i)$ have large variance, i.e., their losses oscillate rapidly between small and large values, and their predictions frequently change, indicating difficulty. The quickly identified easy samples are never unlearnt, and do not suffer from any large loss later in the training. The hard samples are also quickly identified, and remain difficult to learn even after training for many epochs. On average, the dynamics on the hard samples is more consistent with the learning rate schedule, which implies that doing well on these samples can only be achieved at a shared sharp local min-

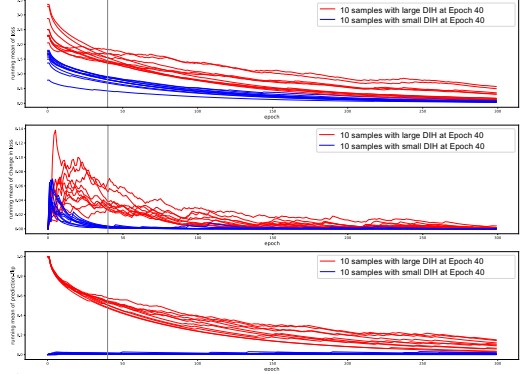

Figure 2: The three strategies (A), (B), and (C) for DIH on ten hard and ten easy samples, each that have been randomly sampled from the top 10k samples with the largest/smallest DIH at Epoch 40.

ima. This suggests that the model generalizes better in the regions containing the easy samples than those containing the hard ones. Similar to human learning, a natural resolution is to learn the hard samples more frequently and reduce the learning on the easy, already learnt, ones. That is, based on $r_t(i)$ (even during the early stages when $t$ is not large), it is prudent to apply additional training effort on hard samples and begin to ignore already learnt easy samples. Further empirical verification can be found in experiments and Figure 4 in the Appendix.

## 2.1 PROBLEM FORMULATION

Figure 2 shows that all three types (base on (A), (B), and (C)) of DIH metrics decrease during training for both easy and hard samples. This indicates that as learning continues, samples become less informative as more training takes place. If we make additional mathematical assumptions, we can model the curriculum learning procedure as a scheme that maximizes an unknown diminishing return function $f(\cdot)$ via only observing its marginal gains $r_t(i)$.

A curriculum is a sequence of selected subsets, where each subset is a mini-batch of data points, i.e. $(S_1, S_2, \ldots, S_T)$. We define a function $f : \mathbb{Z}_{\geq 0}^V \to \mathbb{R}_{\geq 0}$ on the non-negative integer lattice $\mathbb{Z}_{>0}^V$. Each point $z \in \mathbb{Z}_{\geq 0}^V$ is an non-negative integer valued vector of length $|V|$ where $z(i)$ counts

how many times sample $i$ has been selected in the $T$ training data subsets. Any subset $S \subseteq V$ has a characteristic vector $e_S$ where $e_S(i) = 1$ if $i \in S$ and otherwise $s_S(i) = 0$ if $i \notin S$. We also define $S_{1:t} \triangleq \sum_{\ell=1}^{t} e_{S_t}$ as a multi-set input to function $f$ and $S_{1:t} \in \mathbb{Z}_{\geq 0}^{V}$. Ideally, our goal becomes finding the best curriculum $(S_1, S_2, \ldots, S_T)$, i.e., one that maximizes $\bar{f}(S_{1:T})$, as in:

$$\max_{S_{1:T}: \forall t \in [T], S_t \subseteq V, |S_t| \leq k_t} f(S_{1:T}), \tag{2}$$

where $k_t$ is a limit on the size of the set of samples selected at time $t$. However, $f(\cdot)$ can be an arbitrary unknown function in practice. It is also intractable to estimate since it measures the utility of all possible training sequences (in exponential number), so it is inaccessible for optimization. As a surrogate, information about $f(\cdot)$ might be available at each step in the form of the DIH values $r_t(i)$. That is, if we make the mathematical assumption that there is some function $f$ such that $r_t(i) = f(i|S_{1:t-1}) \triangleq f(e_i + S_{1:t-1}) - f(S_{1:t-1})$, then $r_t(i)$ may be used in $f$'s stead and we can optimize the unknown $f(\cdot)$ only based on its partial observation $r_t(i)$. In such case, DIHCL can be seen as a form of online optimization problem whose goal is to find a curriculum that maximizes $f$: at every step $t$, we select a subset of samples (e.g., a mini-batch) $S_t \subseteq V$ of size $|S_t| = k_t$ to train the model, observing only marginal gains $r_t(i)$ of $f(\cdot)$ for each $i$. We can therefore define the following objective:

$$\max_{S_{1:T}: \forall t \in [T], S_t \subseteq V, |S_t| \leq k_t} g(S_{1:T}) \triangleq \sum_{t=1:T} \sum_{i \in S_t} f(i|S_{1:t-1}) \tag{3}$$

For simplicity, we slightly overload the notation of $S_{1:T}$ for function $g(\cdot)$ so that we retain information about the subset selected at every time step (i.e., we can extract $S_t$ for $1 \leq t \leq T$ from $S_{1:T}$). In practice, we update $r_t(i)$ only for samples in $S_t$ since it is a byproduct of training (i.e., the information needed to update $r_t(i)$ requires no more computation than what needed to train the model on $S_t$). Hence, for $i \notin S_t$, $r_t(i) = r_{t-1}(i)$. Although our solution is an approximate solution to the ideal but intractable optimization in Eq. (2), we give its approximation bound in Corollary 1 to the global optimal solution of Eq. (2), and the bound is tight (the best we can get) up to a constant factor.

## 3 DIH CURRICULUM LEARNING

We arrive at a curriculum learning strategy that increases the probability learning on hard samples, and reduces learning on easy ones. However, directly solving Eq. (3) requires costly inference over all the $n$ training samples before selecting every subset $S_t$, as most previous curriculum learning methods do.

### 3.1 A "FREE" CURRICULUM

Most optimization algorithms require inference on the training samples before updating the model parameters, which generates the predictions and losses for the samples used for training. In step $t$, after the model gets trained on $S_t$, the feedback $a_t(i)$ for $i \in S_t$ is already available. However, for $i \notin S_t$, extra inference is inevitable if the curriculum design requires instantaneous instance hardness on the remaining samples to select next subset $S_{t+1}$. By contrast, DIHCL relies on $r_t(i)$ which is a running mean of $a_t(i)$, and

---

**Algorithm 1** DIH Curriculum Learning (DIHCL-Greedy)

1: **input:** $\{(x_i, y_i)\}_{i=1}^{n}, \pi(\cdot; \eta), \eta_{1:T}, \ell(\cdot, \cdot), F(\cdot; w)$;
   $\quad\quad T, T_0; \gamma, \gamma_k \in [0, 1]$
2: **initialize:** $w, \eta_1, k_1 = n, r_0(i) = 1 \; \forall i \in [n]$
3: **for** $t \in \{1, \cdots, T\}$ **do**
4: $\quad$ **if** $t \leq T_0$ **then**
5: $\quad\quad$ $S_t \leftarrow [n]$;
6: $\quad$ **else**
7: $\quad\quad$ Let $S_t = \text{argmax}_{S: |S| = k_t} \sum_{i \in S} r_{t-1}(i)$;
8: $\quad$ **end if**
9: $\quad$ Apply optimization $\pi(\cdot; \eta)$ and record $F(x_i; w_{t-1})$ for $i \in S_t$;

$$w_t \leftarrow w_{t-1} + \pi \left( \nabla_{w_{t-1}} \sum_{i \in S_t} \ell(y_i, F(x_i; w_{t-1})); \eta_t \right)$$

10: $\quad$ Compute normalized $a_t(i)$ for $i \in S_t$ using Eq. (4);
11: $\quad$ Update dynamic instance hardness $r_{t+1}(i)$ using Eq. (1);
12: $\quad$ $k_{t+1} \leftarrow \gamma_k \times k_t$;
13: **end for**

---

it only updates $r_t(i)$ for $i \in S_t$ and keeps $r_t(i)$ for $i \notin S_t$ unchanged, thereby saves extra computation.

At step $t$ of DIHCL, we select subset $S_t \subseteq [n]$ with large $r_{t-1}(i)$ and then update the model by training on $S_t$. We then update $r_t(i)$ via Equation (1). Since the learning rate can change over different steps, and large learning rates means greater model change, we normalize $a_t(i)$ by the learning rate

$\eta_{t-1}$[1]. Specifically, we use one of the following depending on if we're in case (A), (B), or (C):

$$a_t(i) \leftarrow \ell(y_i, F(x_i; w_{t-1}))/\eta_t$$
$$a_t(i) \leftarrow |\ell(y_i, F(x_i; w_{t-1})) - \ell(y_i, F(x_i; w_{\tau_t(i)-1}))|/\sum_{t'=\tau_t(i)}^{t} \eta_{t'} \tag{4}$$
$$a_t(i) \leftarrow \mathbb{1}[\hat{y}_i^{t-1} = \hat{y}_i^{\tau_t(i)-1}]/\sum_{t'=\tau_t(i)}^{t} \eta_{t'}$$

DIHCL is given in Algorithm 1, where $\{(x_i, y_i)\}_{i=1}^n$ is the training data, $\pi(\cdot; \eta)$ is an optimization method such as SGD, $\eta_{1:T}$ are the learning rates of steps 1 to $T$ and $\gamma_k$ is the reduction factor for subset sizes $k_t$. DIHCL trains on more samples early on to produce an initial more accurate estimate of $r_t(i)$. This is indicated by $T_0$, the number of warm start epochs over the whole training set at the start. After this, we start by selecting larger subsets each step and gradually reduce $k_t$ down to the most difficult samples as training proceeds.

A simple method to further reduce training time in the earlier stages is to extract a small subset of $S_t$ by encouraging the diversity of the selected samples. We gradually reduce the diversity preference as training switching to the exploitation stage (reduce $\lambda_t$ by $0 \leq \gamma_\lambda \leq 1$ for every step $t$). Inspired by MCL (Zhou & Bilmes, 2018), after line 7, we reduce $S_t$ to a subset of size $k_t' = \gamma_{k'} k_t$ ($0 < \gamma_{k'} \leq 1$) by (approximately) solving the following submodular maximization.

$$\max_{S \subseteq S_t, |S| \leq k_t'} \sum_{i \in S} r_t(i) + \lambda_t G(S) \tag{5}$$

The function $G : 2^{S_t} \to \mathbb{R}_+$ is may be any submodular function (Fujishige, 2005), and hence we can exploit fast greedy algorithms (Nemhauser et al., 1978; Minoux, 1978; Mirzasoleiman et al., 2015) to solve Eq. (5) with an approximation guarantee.

## 3.2 APPROXIMATION BOUND UNDER FURTHER MATHEMATICAL ASSUMPTIONS

If in addition to the assumption that there exists some function $f : \mathbb{Z}_{\geq 0}^V \to \mathbb{R}_{\geq 0}$ such that $r_t(i) = f(i|S_{1:t-1}) \triangleq f(e_i + S_{1:t-1}) - f(S_{1:t-1})$, we also assume that $f$ has certain properties, then an approximation bound is achievable. The diminishing return (DR) property for $f$ can be defined if, $\forall 0 \leq x \leq y$:

$$f(x + e_i) - f(x) \geq f(y + e_i) - f(y). \tag{6}$$

Recall that $e_i$ is a one-hot vector with all zeros except for a single one at the $i$th position. We also assume $f$ is normalized and monotone, i.e., $f(0) = 0$ and $f(x) \leq f(y), \forall 0 \leq x \leq y$. W.l.o.g. we assume the max singleton gain is bounded by 1 ($\max_i f(e_i) \leq 1$). With such an assumption, we see that $r_t(i)$ is monotonically decreasing with increasing $t$. That is, $r_t(i)$ monotonically decreasing is a necessary, but not sufficient, condition for the DR property on $f$ to hold. Empirically we observe, in Figure 2 that $r_t(i)$ is indeed decreasing, meaning this evidence does not rule out there being a DR function governing $r_t(i)$'s behavior. On the other hand, this of course does not guarantee the DR property. Nevertheless, if it is the case that $r_t(i)$ is produced as above from some DR function, it enables the following analysis to proceed.

Under the above assumptions, we may derive bounds of DIHCL-Greedy(Alg. 1) when $k_t = k$ $\forall t \in [T]$. For simplicity, assume $n \mod k = 0$ and let $m \triangleq \frac{n}{k}$. We first show the bound on function $g$ of observed gains, and then connect it to the unknown function $f$.

**Theorem 1.** *For $f : \mathbb{Z}_{\geq 0}^V \to \mathbb{R}_{\geq 0}$ on ground set $V$ with DR property, compared to any solution $S_{1:T}^*$, $S_{1:T}$, the solution of DIHCL-Greedy, achieves*

$$g(S_{1:T}) + C_{f,m} \geq \max\left\{\frac{1 - e^{-1}}{k}, \frac{k}{2n}\right\} g(S_{1:T}^*), \tag{7}$$

*Where $C_{f,m} \triangleq m \min_{A_{1:m}} g(A_{1:m})$ such that $\bigcup_{i=1}^m A_i = V$, and $|A_i| = k$.*

**Corollary 1.** $f(S_{1:T}) + \frac{1}{k}C_{f,m} \geq \frac{1}{k}\max\left\{\frac{1-e^{-1}}{k}, \frac{k}{2n}\right\} f(S_{1:T}^*)$

The proofs are given the Appendix. The $C_{f,m}$ term in the bound reflects our loss during the warm start phase, where we cannot estimate the gain of each sample unless we select each sample at least once, which is independent of $T$ and vanishes in the long run. The $1 - e^{-1}$ comes from the DR property and our greedy procedure. For the $1/k$ factor and the $k/n$ factor of the bound on $g$, we give hard cases in the Appendix so our bound is tight to constant factors. These factors result from

---

[1] We use $\eta_{t-1}$ instead of $\eta_t$ because $a_t(i)$ is computed based on $w_{t-1}$ before the weight update in step $t$.

our assumption about the function $f$, which may have arbitrary interactions among data points. In practice, similar data points tend to have similar DIH, and we can incorporate such information by adding an additional term of submodular function $G$ to the DIH value to model data point interactions.

## 3.3 PRACTICAL METHODS FOR DIH-WEIGHTED SAMPLING IN DIHCL

In line 7 of Alg. 1, we select $S_t$ with the highest $r_{t-1}(i)$ values. In practice, we find adding randomness to the selection procedure gives better performance as (1) exploration on samples with small $r_t(i)$ is necessary for accurate estimate to $r_t(i)$, and (2) randomness of training samples is essential to achieve a good quality solution $w$ for non-convex models such as DNNs. Instead of choosing greedily the top $k_t$ samples, we perform random sampling with probability $p_{t,i} \propto h(r_{t-1}(i))$, where $h(\cdot)$ is a monotone non-decreasing function, and we still prefer data points with high DIH. An ideal choice of $h(\cdot)$ should balance between the exploration of data with poorly estimated DIH and exploitation of data with well estimated DIH. We propose the following three sampling methods to replace line 7 of Alg. 1, and give extensive evaluations in the experiment section.

**DIHCL-Rand:** Let $h(r_t(i)) = r_t(i)$. We sample data points weighted by their DIH values.

**DIHCL-Exp:** We trade-off exploration and exploitation similarly to Exp3 (Auer et al., 2003), which samples based on the softmax value and reweigh the observation by the selection probability to encourage exploration:

$$h(r_t(i)) = \exp\left[\sqrt{2\log n/n} \times r_t(i)\right], \quad a_t(i) \leftarrow a_t(i)/p_{t,i} \ \forall i \in S_t. \tag{8}$$

**DIHCL-Beta:** We utilize the idea of Thompson sampling (Thompson, 1933) and use a Beta distribution prior to balance exploration and exploitation, i.e., $h(r_t(i)) \sim \text{Beta}(r_t(i), c - r_t(i))$, where $c$ is a sufficiently large constant that $c \geq r_t(i)$, e.g., $c = 1$ when $a_t(i)$ is prediction flip. The Beta distribution encourages exploration when the difference between $r_t(i)$ and $c - r_t(i)$ is small.

Table 1: The final test accuracy (%) achieved by different methods training DNNs on 11 datasets (without pre-training). We use "Loss, dLoss, Flip" to respectively denote the 3 choices of DIH metrics based on (A), (B), and (C) respectively. In all DIHCL variants, we apply lazier-than-lazy-greedy (Mirzasoleiman et al., 2015) for Eq. (5) on all datasets except Food-101, Birdsnap, Aircraft (FGVC Aircraft), Cars (Stanford Cars), and ImageNet.

| Curriculum | CIFAR10 | CIFAR100 | Food-101 | ImageNet | STL10 | SVHN | KMNIST | FMNIST | Birdsnap | Aircraft | Cars |
|---|---|---|---|---|---|---|---|---|---|---|---|
| Rand mini-batch | 96.18 | 79.64 | 83.56 | 75.04 | 86.06 | 96.48 | 98.67 | 95.22 | 64.23 | 74.71 | 78.73 |
| SPL | 93.55 | 80.25 | 81.36 | 73.23 | 81.33 | 96.15 | 97.24 | 92.09 | 63.26 | 68.95 | 77.61 |
| MCL | 96.60 | 80.99 | 84.18 | 75.09 | 88.57 | 96.93 | 99.09 | 95.07 | 65.76 | 75.28 | 76.98 |
| DIHCL-Rand, Loss | 96.76 | 80.77 | 83.82 | 75.41 | 87.25 | 96.81 | 99.10 | **95.69** | 65.62 | 79.00 | 80.91 |
| DIHCL-Rand, dLoss | 96.73 | 80.65 | 83.82 | 75.34 | 86.93 | 96.83 | 99.14 | 95.64 | 65.25 | **79.93** | 78.70 |
| DIHCL-Exp, Loss | **97.03** | **82.23** | 84.65 | 75.10 | 88.36 | 96.91 | **99.20** | 95.45 | 66.13 | 77.68 | 79.85 |
| DIHCL-Exp, dLoss | 96.40 | 81.42 | 84.75 | 75.62 | **89.41** | 96.80 | 99.18 | 95.50 | **66.59** | 79.72 | **81.48** |
| DIHCL-Beta, Flip | 96.51 | 81.06 | **84.94** | **76.33** | 86.88 | **97.18** | 99.05 | 95.66 | 65.48 | 78.49 | 80.13 |

## 4 EMPIRICAL EXPERIMENTAL EVALUATION

We train different DNNs by using variants of DIHCL, and compare them with three baselines, vanilla random mini-batch SGD, self-paced learning (SPL) (Kumar et al., 2010), and minimax curriculum learning (MCL) (Zhou & Bilmes, 2018) on 11 image classification datasets (without pre-training), i.e., (1) WideResNet-28-10 (Zagoruyko & Komodakis, 2016) on CIFAR10 and CIFAR100 (Krizhevsky & Hinton, 2009); (2) ResNeXt50-32x4d (Xie et al., 2017) on Food-101 (Bossard et al., 2014), FGVC Aircraft (Aircraft) (Maji et al., 2013), Stanford Cars (Krause et al., 2013), and Birdsnap (Berg et al., 2014); (3) ResNet50 He et al. (2016) on ImageNet (Deng et al., 2009); (4) WideResNet-16-8 on Fashion-MNIST (FMNIST) (Xiao et al., 2017) and Kuzushiji-MNIST (KMNIST) (Clanuwat et al., 2018); (5) PreActResNet34 He et al. (2016) on STL10 (Coates et al., 2011) and SVHN (Netzer et al., 2011). We use mini-batch SGD with momentum of 0.9 and cyclic cosine annealing learning rate schedule (Loshchilov & Hutter, 2017) (multiple episodes with starting/target learning rate decayed by a multiplicative factor 0.85). We use $T_0 = 5, \gamma = 0.95, \gamma_k = 0.85$ for all DIHCL variants, and gradually reduce $k$ from $n$ to $0.2n$. On each dataset, we apply each method to train the same model for the same number of epochs, but each method may select different amount of samples per epoch. More details about the datasets and settings can be found in the Appendix. For DIHCL variants that further reduce $S_t$ by solving Eq. (5), we use $\lambda_1 = 1.0, \gamma_\lambda = 0.8, \gamma_{k'} = 0.4$ and employ the "facility location"

submodular function (Cornuéjols et al., 1977) $G(S) = \sum_{j \in S_t} \max_{i \in S} \omega_{i,j}$ where $\omega_{i,j}$ represents the similarity between sample $x_i$ and $x_j$. We utilize a Gaussian kernel for similarity using neural net features (e.g., the inputs to the last fully connected layer in our experiments) $z(x)$ for each $x$, i.e., $\omega_{i,j} = \exp\left(-\|z(x_i) - z(x_j)\|^2 / 2\sigma^2\right)$, where $\sigma$ is the mean value of all the $k(k-1)/2$ pairwise distances.

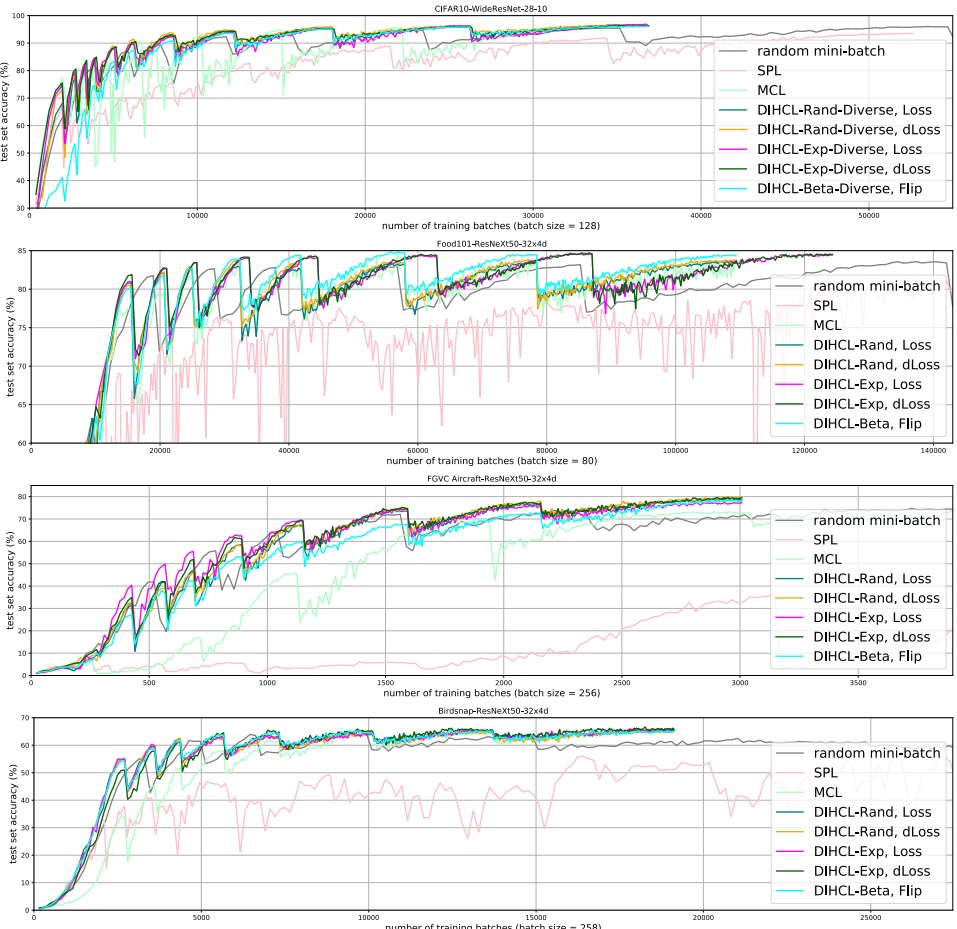

Figure 3: Training DNNs by using DIHCL (and its variants), SPL (Kumar et al., 2010), MCL (Zhou & Bilmes, 2018), and random mini-batch SGD on CIFAR10, Food-101, FGVC Aircraft and Birdsnap. We use "Diverse" to denote DIHCL that further reduces $S_t$ by applying submodular maximization for Eq. (5). We report how the test accuracy changes with the number of training batches for each method.

In Figure 3, we show how the test set accuracy changes when increasing the number of training batches in each curriculum learning method on 3 datasets. The results for other 8 datasets can be found in the Appendix, together with the wall-clock time for (1) the entire training and (2) the submodular maximization part in DIHCL with diversity and MCL. The final test accuracy achieved by each method is reported in Table 1. DIHCL and its variants show significantly faster and smoother gains on test accuracy than baselines during training especially at earlier stages. They also achieve higher final accuracy and show improvements in sample efficiency (meaning they reach their best performance sooner, after less computation has taken place). MCL can reach similar performance as DIHCL on some datasets but it shows less stability and requires more computation for submodular maximization. We also observe a similar instability of SPL. The reason is that, compared to the methods that use DIH, both MCL and SPL deploy instantaneous instance hardness (i.e., current loss) as the score to select samples, a measure that is more sensitive to randomness and perturbation that occurs during training. Compared to MCL and DIHCL, SPL and the random mini-batch curriculum method requires more epochs to reach their best accuracy, since they spend training effort on the easier and memorable samples but lack sufficient repeated-learning of the forgettable ones. Although every variant of DIHCL achieves the best accuracy among all the evaluated methods on some datasets, DIHCL-Exp using loss and DIHCL-Beta using prediction flip, as the instantaneous hardness, exhibit advantages over the other DIHCL variants. One possible explanation is that the running mean computed on the

loss and prediction flips are more stable along the training trajectory as shown in Figure 2, or perhaps they are more in line with our assumption in Section 3.2 about the diminishing return property of $f(\cdot)$.

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

# A    PROOFS

$f : \mathbb{Z}_{\geq 0}^V \to \mathbb{R}_{\geq 0}$ on ground set $V$ is defined over an integer lattice. The diminishing return (DR) property of $f$ is the following inequality $0 \leq \forall x \leq y$:

$$f(x + e_i) - f(x) \geq f(y + e_i) - f(y), \tag{9}$$

Where $e_i$ is a one-hot vector with all zeros except for a single one at the $i$th position. We assume $f$ is normalized and monotone, i.e., $f(0) = 0$ and $f(x) \leq f(y), \forall 0 \leq x \leq y$. W.l.o.g. we also assume the max singleton gain is bounded by 1, i.e., $\max_i f(e_i) \leq 1$. We can think that $f$ takes input as a multi-set, and the gain of an item diminishes as its counter increases in the multi-set.

In the setting of selecting mini-batches for training machine learning models, suppose the mini-batch size is $k$, the training set is $V$, and at every time step $t$, we select $S_t \subseteq V$ with $|S_t| = k$, and only observe the gains on the selected subset (e.g., for neural networks, we update the running mean of training losses during the forward pass of the chosen mini-batch, or DIH type (A)). At every time step of selecting a mini-batch, we observe $f(i|S_{1:t-1}) \forall i \in S_t$. Let $n = |V|$, $m = \frac{n}{k}$, and for simplicity assume $n \mod k = 0$. We define function $g$ to reflect the observed gains from $f$ as we select data samples at each training step:

$$g(S_{1:t}) = \sum_{t'=1:t} \sum_{i \in S_{t'}} f(i|S_{1:t'-1}) \tag{10}$$

For simplicity, we slightly overload the notation of $S_{1:T}$ for function $g(\cdot)$ so that we retain information about the subset selected at every time step (i.e., we can extract $S_t$ for $1 \leq t \leq T$ from $S_{1:T}$). Note that $g$ is permutation-variant for $k > 1$, i.e., for different ordering in $S_{1:t}$, $g$ gives different values.

**Theorem 1.** *For $f : \mathbb{Z}_{\geq 0}^V \to \mathbb{R}_{\geq 0}$ on ground set $V$ with DR property, compared to any solution $S_{1:T}^*$, $S_{1:T}$, the solution of* DIHCL-Greedy*, achieves*

$$g(S_{1:T}) + C_{f,m} \geq \max \left\{ \frac{1 - e^{-1}}{k}, \frac{k}{2n} \right\} g(S_{1:T}^*), \tag{7}$$

*Where $C_{f,m} \triangleq m \min_{A_{1:m}} g(A_{1:m})$ such that $\bigcup_{i=1}^m A_i = V$, and $|A_i| = k$.*

To bridge $S_{1:T}$ with $S_{1:T}^*$, we first connect $S_{1:T}$ to the greedy solution with singleton gain oracle, but uses the history of sequence of $(S_1, S_2, \ldots, S_{T-1})$, which we denote by $(\hat{S}_1, \hat{S}_2, \ldots, \hat{S}_T)$:

$$\hat{S}_t = \operatorname*{argmax}_{S \subseteq V, |S|=k} \sum_{i \in S} f(i|S_{1:t-1}). \tag{11}$$

Note we denote any set with subscript 0 (at time step 0) as an empty set, i.e. $S_0 = \emptyset$, $\hat{S}_0 = \emptyset$, and etc.. We define the observed gain values on the singleton gain oracle with history of $(S_1, S_2, \ldots, S_{T-1})$ as:

$$g(\hat{S}_{1:T}|S_{1:T-1}) = \sum_{t=1:T} \sum_{i \in \hat{S}_t} f(i|S_{1:t-1}) \tag{12}$$

Firstly, we derive a lower bound of $g(S_{1:T})$ in terms of $g(\hat{S}_{1:T}|S_{1:T-1})$.

**Lemma 1.** $g(S_{1:T}) + C_{f,m} \geq g(\hat{S}_{1:T}|S_{1:T-1})$.

*Proof.* Define $\zeta(i, A_{1:t})$ to return the subsequence of $A_{1:t}$ that starts from $A_1$ and ends at $A_{t'}$ where $A_{t'}$ is the last set in the whole sequence that contains the element $i$, i.e., $\zeta(i, A_{1:t}) = \operatorname*{argmax}_{A_{1:t'}} t' \mathbb{1}_{i \in A_{t'}}$. When $i$ is not present in the whole sequence $A_{1:t}$, $\zeta(i, A_{1:t})$ returns $\emptyset$.

By definitions of $C_{f,m}$ and $g(\hat{S}_{1:m}|S_{1:m-1})$, we have $C_{f,m} \geq mf(V) \geq g(\hat{S}_{1:m}|S_{1:m-1})$ due to the diminishing return (DR) property.

For $T \leq m$, Lemma 1 is true because of the above inequality.

For $T \geq m + 1$, we compare the previous gains of elements in $S_t$ to the current gains of elements in $\hat{S}_t$:

$$g(S_{1:T}) + C_{f,m} \geq g(S_{1:T}) + g(\hat{S}_{1:m}|S_{1:m-1}) \tag{13}$$

$$\geq \sum_{t=m+1:T} \sum_{i \in S_t} f(i|\zeta(i, S_{1:t-1})) + g(\hat{S}_{1:m}|S_{1:m-1}) \tag{14}$$

$$\geq \sum_{t=m+1:T} \sum_{i \in \hat{S}_t} f(i|\zeta(i, S_{1:t-1})) + g(\hat{S}_{1:m}|S_{1:m-1}) \tag{15}$$

$$\geq \sum_{t=m+1:T} \sum_{i \in \hat{S}_t} f(i|S_{1:t-1}) + g(\hat{S}_{1:m}|S_{1:m-1}) \tag{16}$$

$$= g(\hat{S}_{1:T}|S_{1:T-1}) \tag{17}$$

Eq. 13 and Eq. 16 hold due to the diminishing return property and Eq. 15 is a result of the greedy step (i.e., $S_t$ is optimal when conditioning on $\zeta(i, S_{1:t-1})$). Note that we are guaranteed to find an element in the sequence history ($|\zeta(i, S_{1:t-1})| > 0$ in Eq. 14 and Eq. 15) since we sweep the ground set $V$ in the first $m$ steps of solution $S_{1:m}$.

$\square$

*Remarks.* In the proof, we ignore the gain at the $T$ step, i.e., $\sum_{i \in S_T} f(i|S_{1:T-1})$ as such gain can potentially be zero. In other words, $g(S_{1:T-1}) + C_{f,m} \geq g(\hat{S}_{1:T}|S_{1:T-1})$. For the case that $f$ is modular, i.e., $f(x + e_i) = f(x) + f(e_i)$, and for only $k$ elements in $V$, the function evaluations are non-zero, the bound meets in equality: $g(S_{1:T-1}) + C_{f,m} = g(\hat{S}_{1:T}|S_{1:T-1})$. The idea is that we have to sweep all elements in the ground set before we identify the non-zero-valued elements.

Next, we find a lower bound of $g(\hat{S}_{1:T}|S_{1:T-1})$ in terms of $g(S_{1:T}^*)$.

**Lemma 2.** $g(\hat{S}_{1:T}|S_{1:T-1}) \geq \frac{1-e^{-1}}{k} g(S_{1:T}^*)$.

*Proof.* For $T' < T$, we compare $g(S_{1:T}^*)$ with $g(\hat{S}_{1:T}|S_{1:T-1})$:

$$\frac{1}{k} g(S_{1:T}^*) = \frac{1}{k} \sum_{t=1:T} \sum_{i \in S_t^*} f(i|S_{1:t-1}^*) \tag{18}$$

$$\leq \frac{1}{k} \sum_{t=1:T} k \times \max_{i \in S_t^*} f(i|S_{1:t-1}^*) \tag{19}$$

$$\leq \frac{1}{k} k f(S_{1:T}^*) \tag{20}$$

$$\leq f(S_{1:T'}^* + S_{1:T}) \tag{21}$$

$$\leq f(S_{1:T'}) + \sum_{i \in S_{1:T}^*} f(i|S_{1:T'}) \tag{22}$$

$$\leq f(S_{1:T'}) + T(\sum_{i \in \hat{S}_{T'+1}} f(i|S_{1:T'})) \tag{23}$$

$$= f(S_{1:T'}) + T(g(\hat{S}_{1:T'+1}|S_{1:T'}) - g(\hat{S}_{1:T'}|S_{1:T'-1})) \tag{24}$$

$$\leq g(\hat{S}_{1:T'}|S_{1:T'-1}) + T(g(\hat{S}_{1:T'+1}|S_{1:T'}) - g(\hat{S}_{1:T'}|S_{1:T'-1})) \tag{25}$$

From Eq. 19 to Eq. 20, we use $\sum_{t=1:T} \max_{i \in S_t^*} f(i|S_{1:t-1}^*) \leq \sum_{t=1:T} f(S_t^*|S_{1:t-1}^*) = f(S_{1:T}^*)$.

Eq. 22 is due to DR property and Eq. 23 is a result of greedy selection. Also note that for Eq. 21, $S_{1:T'}^* + S_{1:T} = \sum_{l=1}^{T'} e_{S_l^*} + \sum_{l=1}^{T} e_{S_l}$.

By rearranging Eq. 25, we have $\frac{1}{T}(\frac{1}{k} g(S_{1:T}^*) - g(\hat{S}_{1:T'}|S_{1:T'-1}))) \leq g(g(\hat{S}_{1:T'+1}|S_{1:T'}) - g(\hat{S}_{1:T'}|S_{1:T'-1})$, i.e., every time step, we reduce the gap to $1/k$ of the of optimal solution by at least $\frac{1}{T}$. Therefore $g(\hat{S}_{1:T}|S_{1:T-1}) \geq \frac{1-e^{-1}}{k} g(S_{1:T}^*)$. $\square$

*Remarks.* We will show that there is a hard case with $1/k$ factor. Suppose $f$ is a set cover function ($f(i|A) = 0$ if $i \in A$) and $|V| = k^2$. The ground set $V$ is partitioned into $k$ groups $V = V_1 \cup V_2 \cup \ldots \cup V_k$ with $k$ elements in each group, such that $f(a) = 1 \, \forall a \in V$, $f(a|b) = 0 \, \forall a, b \in V_i$, and $f(\{a, b\}) = f(a) + f(b) \, \forall a \in V_i, b \in V_j, i \neq j$. For the first time step, $g(\hat{S}_1|\emptyset)$ gets a gain of $k$ which is equal to $g(S_1^*)$. However, $S_1$ may select one element from each of the group since we are doing the ground set sweeping exploration, and all the rest gains will be zero conditioned on $S_1$. The optimal solution, on the other hand, can select all $k$ elements from one group at a time, and get a value of $k^2$ in the end.

Combine Lemma 1 and Lemma 2, we get the first factor $\frac{1-e^{-1}}{k}$ for the bound in Theorem 1.

**Lemma 3.** $g(\hat{S}_{1:T}|S_{1:T-1}) \geq \frac{k}{2n} g(S_{1:T}^*)$.

*Proof.* We will first connect $g(\hat{S}_{1:T}|S_{1:T-1})$ with the solution that selects the entire ground set $V$ at every step, i.e., $g(V_{1:T}) = g((V, V, \ldots, V))$.

$$g(\hat{S}_{1:T}|S_{1:T-1}) \geq \frac{k}{n} g(V_{1:T}|S_{1:T-1}) \tag{26}$$

$$\geq \frac{k}{n} g(V_{1:T}|V_{1:T-1}) \tag{27}$$

$$= \frac{k}{n} \sum_{t=1:T} \sum_{i \in V} f(i|V_{1:t-1}) \tag{28}$$

$$\geq \frac{k}{n} f(V_{1:T}) \tag{29}$$

For Eq. 26, we use the fact that $\hat{S}_{1:T}$ achieve the top $k$ gains selected by the greedy process in each step. Next, we will bound any solution $g(S_{1:T}^*)$ by $g(V_{1:T}|V_{1:T-1})$. Firstly, we will need to partition $S_{1:T}^*$ into two parts: (1) for the first part, we collect all the new elements introduced at every time step $t$ that do not exist in $S_{1:t-1}^*$, i.e., $\tilde{S}_{1:T}^* = (S_1^* \setminus \emptyset, S_2^* \setminus \cup(S_{1:1}^*), S_3^* \setminus \cup(S_{1:2}^*), \ldots, S_T^* \setminus \cup(S_{1:T-1}^*))$, where $\tilde{S}_t^* \triangleq S_t^* \setminus \cup(S_{1:t-1}^*)$ and $\cup(S_{1:t}) \triangleq \bigcup_{i=1:t} S_i$, which is the set union on all elements in the multiset (you can think it sets all the counters in the multiset with values $\geq 1$ to ones), and "\" is the set minus operation. Therefore, $\tilde{S}_{1:T}^*$ contains every element in $S_{1:T}^*$ exactly once, i.e., every element in $S_{1:T}^*$ only appears once in $\tilde{S}_{1:T}^*$, and at many time steps, $\tilde{S}_t^*$ might be empty; (2) the other part contains all the rest elements, i.e., $S_{1:T}^* - \tilde{S}_{1:T}^* = (S_1^* \setminus \tilde{S}_1^*, S_2^* \setminus \tilde{S}_2^*, \ldots, S_T^* \setminus \tilde{S}_T^*)$. We bound the two parts as follows:

$$g(S_{1:T}^*) = \sum_{t=1:T} \sum_{i \in S_t^*} f(i|S_{1:t-1}^*) \tag{30}$$

$$= \sum_{t=1:T} \sum_{i \in \tilde{S}_t^*} f(i|S_{1:t-1}^*) + \sum_{t=1:T} \sum_{i \in (S_t^* \setminus \tilde{S}_t^*)} f(i|S_{1:t-1}^*) \tag{31}$$

$$\leq \sum_{i \in V} f(i) + \sum_{t=1:T} \sum_{i \in (S_t^* \setminus \tilde{S}_t^*)} f(i|S_{1:t-1}^*) \tag{32}$$

$$\leq \sum_{i \in V} f(i) + f(S_{1:T}^*) \tag{33}$$

$$\leq g(V_{1:T}|V_{1:T-1}) + f(V_{1:T}) \tag{34}$$

From Eq. 31 to Eq. 32, we use the fact $\sum_{t=1:T} \sum_{i \in \tilde{S}_t^*} f(i|S_{1:t-1}^*) \leq \sum_{t=1:T} \sum_{i \in \tilde{S}_t^*} f(i) \leq \sum_{i \in V} f(i)$ since $\tilde{S}_T^*$ contains one instance of every element in $S_{1:T}^*$ and removing the conditioning part would make the gains larger (guaranteed by diminishing return property). To get Eq. 33, we reduce the conditioning part of $f(i|S_{1:t-1}^*)$ in Eq. 32 by using the following inequality: for $A_1 \subseteq A_2 \subseteq V$, denote $A_3 = A_2 \setminus A_1$ and let $A_1 = \{i_1, i_2, \cdots, i_{|A_1|}\}$, by diminishing return property of $f(\cdot)$, we have:

$$\sum_{i \in A_1} f(i|A_2) \leq f(i_1|A_3) + f(i_2|\{i_1\} \cup A_3) + f(i_3|\{i_1, i_2\} \cup A_3) +$$

$$\ldots + f(i_{|A_1|}|A_2 \setminus \{i_{|A_1|}\}) = f(A_2|A_3). \tag{35}$$

According to the pre-defined partition, we pick out the first occurrence of every element into $\tilde{S}^*_{1:T}$, every remaining element $i \in (S^*_t \setminus \tilde{S}^*_t)$ is guaranteed to find itself in its conditioning history $S^*_{1:t-1}$ and therefore, we may use the inequality described in Eq. 35 to bound the second term in Eq. 32 by $f(S^*_{1:T})$ (letting $A_1 = S^*_t \setminus \tilde{S}^*_t$ and $A_2 = S^*_{1:t-1}$ and applying the inequality from $t = 1$ to $T$ sequentially). To make it more concrete, for example, at step $t = 2$, by using Eq. 35, we have:

$$\sum_{i \in (S^*_2 \setminus \tilde{S}^*_2)} f(i|S^*_1) \le f(i_1) + f(i_2|i_1) + f(i_3|i_1, i_2)$$

$$+ \ldots + f(i_{|S^*_2 \setminus \tilde{S}^*_2|}|S^*_2 \setminus \tilde{S}^*_2 \setminus \{i_{|S^*_2 \setminus \tilde{S}^*_2|}\}) \tag{36}$$

$$= f(S^*_2 \setminus \tilde{S}^*_2); \tag{37}$$

At time step $t = 3$, we have:

$$\sum_{i \in (S^*_3 \setminus \tilde{S}^*_3)} f(i|S^*_{1:2}) \le f(i_1|S^*_2 \setminus \tilde{S}^*_2) + f(i_2|\{i_1\} \cup (S^*_2 \setminus \tilde{S}^*_2)) + f(i_3|\{i_1, i_2\} \cup (S^*_2 \setminus \tilde{S}^*_2))$$

$$+ \ldots + f(i_{|S^*_3 \setminus \tilde{S}^*_3|}|(S^*_3 \setminus \tilde{S}^*_2 \setminus \{i_{|S^*_3 \setminus \tilde{S}^*_3|}\}) \cup (S^*_2 \setminus \tilde{S}^*_2)) \tag{38}$$

$$= f(S^*_3 \setminus \tilde{S}^*_3|S^*_2 \setminus \tilde{S}^*_2). \tag{39}$$

Hence, we have the inequality between Eq. 32 and Eq. 33.

To get Eq. 34 from Eq. 33, we use the fact $\sum_{i \in V} f(i) \le g(V_{1:T}|V_{1:T-1})$ because $g(V_{1:T}|V_{1:T-1})$ contains $\sum_{i \in V} f(i)$ at step $t = 1$, and the second term in Eq. 34 is due to the fact that $f(\cdot)$ is monotone non-decreasing.

Finally, we combine Eq. 29 and Eq. 34, we get $2g(\hat{S}_{1:T}|S_{1:T-1}) \ge \frac{2k}{n} f(V_{1:T}) \ge \frac{k}{n} g(S^*_{1:T})$.

$\square$

By combining Lemma 1 and Lemma 3, we get the second factor $\frac{k}{2n}$ for the bound in Theorem 1.

*Remarks.* The first factor $\frac{1-e^{-1}}{k}$ dominates when $k$ is relatively small compared to $n$. Recall the hard case example above on the $1/k$ factor. We can generalize it to any $k < n$ by (almost) equally distribute the $n$ elements into the $k$ groups described in the hard case. Then, for $n < k^2$, the optimal solution gets $n$ in the end while the greedy solution gets $k$, so the ratio is $\frac{k}{n}$. For $n \ge k^2$, the optimal solution still gets $k^2$ while the greedy solution gets $k$, so the ratio is $\frac{1}{k}$. In both scenarios, our bounds match the hard example up to constant factors.

**Corollary 1.** $f(S_{1:T}) + \frac{1}{k} C_{f,m} \ge \frac{1}{k} \max \left\{ \frac{1-e^{-1}}{k}, \frac{k}{2n} \right\} f(S^*_{1:T})$

*Proof.*

$$f(S_{1:T}) + \frac{1}{k} C_{f,m} \ge \frac{1}{k} (g(S_{1:T}) + C_{f,m}) \tag{40}$$

$$\ge \frac{1}{k} \max\{\frac{1-e^{-1}}{k}, \frac{k}{2n}\} g(S^*_{1:T}) \tag{41}$$

$$\ge \frac{1}{k} \max\{\frac{1-e^{-1}}{k}, \frac{k}{2n}\} f(S^*_{1:T}) \tag{42}$$

$\square$

*Remarks.* We will show there is a hard case with the $1/k^2$ factor. The same as the hard case mentioned above for the set cover function, $f(\hat{S}_1)$ gets a gain of 1 since the selected items can be totally redundant, and the future gains are all zeros since $S_1$ select one element from each group. However, the optimal solution can still achieve an evaluation of $k^2$ in the end. Also, note that Theorem 1 is true for any solution $S^*_{1:T}$ and the optimal solution for $g$ and the optimal solution for $f$ can be different.

We mentioned a few weighted sampling method to replace the greedy step. Here, we apply a random sampling procedure similar to the lazier-than-lazy approach Mirzasoleiman et al. (2015): we sample a subset $R_j \subseteq V \setminus S_{t,j-1}$ of size $\frac{n}{k} \log \frac{1}{\epsilon}$, and then choose the top-gain element from $R_j$ and add it to $S_{t,j-1}$ to from $S_{t,j}$. We denote such sampling based greedy as DIHCL-Greedy-random.

**Theorem 2.** *For $f : \mathbb{Z}_{\geq 0}^V \to \mathbb{R}_{\geq 0}$ on ground set $V$ with DR property, compared to any solution $S_{1:T}^*$, $S_{1:T}$, the solution of DIHCL-Greedy-random, achieves*

$$\mathbb{E}[g(S_{1:T})] + C_{f,m} \geq (1 - (1 - \frac{1-\epsilon}{k})^k)\frac{1-e^{-1}}{k}g(S_{1:T}^*) \tag{43}$$

$$\geq \frac{(1-e^{-1}-\epsilon)(1-e^{-1})}{k}g(S_{1:T}^*). \tag{44}$$

*Proof.* We can think the selection of every $S_t$ is a greedy process of $k$ steps, with $S_t$ as the optimal solution. Suppose up to step $j$, we select the set $S_{t,j}$. We first bound the probability that the sampled set has some intersection with the optimal set $S_t$.

$$\Pr[R_j \cap (S_t \setminus S_{t,j-1}) \neq \emptyset] \geq 1 - (1 - \frac{|S_t \setminus S_{t,j-1}|}{|V \setminus S_{t,j-1}|})^{|R|} \tag{45}$$

$$\geq 1 - (1 - \frac{|S_t \setminus S_{t,j-1}|}{n})^{|R|} \tag{46}$$

$$\geq 1 - e^{-\frac{|R|}{n}|S_t \setminus S_{t,j-1}|} \tag{47}$$

$$\geq (1 - e^{-\frac{|R|k}{n}})\frac{|S_t \setminus S_{t,j-1}|}{k} \tag{48}$$

$$\tag{49}$$

In step $j$, we denote the selected item by $v_j$. We can then get the expected gain given the probability that there is some intersection:

$$\mathbb{E}[f(v_j|\zeta(v_j, S_{1:t-1}))] \geq \Pr[R_j \cap (S_t \setminus S_{t,j}) \neq \emptyset]\frac{1}{|S_t \setminus S_{t,j}|}\sum_{i \in S_t} f(i|\zeta(i, S_{1:t-1})) \tag{50}$$

$$= \frac{1-\epsilon}{k}\sum_{i \in S_t} f(i|\zeta(i, S_{1:t-1})) \tag{51}$$

Again, we get the argument that we are reducing the gap to the optimal solution by $(1-\epsilon)/k$ for every selected item $v_j$ on expectation.

$$\sum_{j=1:k} \mathbb{E}[f(v_j|\zeta(v_j, S_{1:t-1}))] \geq (1 - (1 - \frac{1-\epsilon}{k})^k)\sum_{i \in S_t} \mathbb{E}[f(i|\zeta(i, S_{1:t-1}))] \tag{52}$$

We can then apply Eq. 52 in the Eq. 14 of Lemma 1, and get

$$\mathbb{E}[g(S_{1:T})] + C_{f,m} \geq (1 - (1 - \frac{1-\epsilon}{k})^k)\mathbb{E}[g(\hat{S}_{1:T}|S_{1:T-1})] \tag{53}$$

Combine with Lemma 2 we get the bound in Theorem 2.

$\square$

*Remarks.* When $n$ is large and $n \gg k$, we can approximate the sample without replacement using sample with replacement, and we can independently sample $k$ subsets each of size $|R|$ at every time step to generate $S_k$. In such a case, the bound becomes $\mathbb{E}[g(S_{1:T})] + C_{f,m} \geq \frac{1-e^{-1}-\epsilon}{k}g(S_{1:T}^*)$.

Similarly, we can also get the expectation bound on $f$:

**Corollary 2.** $\mathbb{E}[f(S_{1:T})] + \frac{1}{k}C_{f,m} \geq (1 - (1 - \frac{1-\epsilon}{k})^k)\frac{1-e^{-1}}{k^2}f(S_{1:T}^*)$

*Proof.*

$$\mathbb{E}[f(S_{1:T})] + \frac{1}{k}C_{f,m} \geq \frac{1}{k}(\mathbb{E}[g(S_{1:T})] + C_{f,m}) \tag{54}$$

$$\geq (1 - (1 - \frac{1-\epsilon}{k})^k)\frac{1-e^{-1}}{k^2}g(S_{1:T}^*) \tag{55}$$

$$\geq (1 - (1 - \frac{1-\epsilon}{k})^k)\frac{1-e^{-1}}{k^2}f(S_{1:T}^*) \tag{56}$$

$\square$

We can extend the setting so that we get noisy feedback from the gains of function $f$: $f(i|S_{1:t-1})+\alpha_t$, and the problem becomes a multi-armed bandit problem. Specifically if we assume the noise $a_t$ form a martingale difference sequence, i.e. $\mathbb{E}[\alpha_t|\alpha_1, \alpha_2, \ldots, \alpha_{t-1}] = 0$ and all $\alpha_t$ are bounded $\alpha_t \leq \sigma$ and if we make further assumption about the smoothness of the $f$ and $g$ function (assume the gains of $f$ and $g$ have RKHS-norm bounded by value $B$ with some kernel $\mathbf{k}$), we can utilize the contextual bandit UCB algorithm proposed in Krause & Ong (2011) to get a $\sqrt{T}$ dependent regret. Also, under the noise setting, the contextual information becomes crucial, as the function has DR-property, and without an estimate of how much the gain decreases, we cannot have a better estimate of the upper bound on the noise term. However, we note that utilizing the contextual information involves calculating large kernel matrices, which is not feasible for our purpose of efficient curriculum learning. We include the following result for completeness.

**Theorem 3.** *For $f : \mathbb{Z}_{\geq 0}^V \to \mathbb{R}_{\geq 0}$ on ground set $V$ with DR property, suppose the gain of function $g$ has RKHS-norm bounded by value $B$ with some kernel $\mathbf{k}$), and the noise $\alpha_t$'s from a martingale difference sequence: $\mathbb{E}[\alpha_t|\alpha_1, \alpha_2, \ldots, \alpha_{t-1}] = 0$ and all $\alpha_t$ are bounded $|\alpha_t| \leq \sigma$. We define the maximum information gain if we have the perfect information about $f$, $\rho_T = \max_{A_{1:T}} H(y_{A_{1:T}}) - H(y_{A_{1:T}}|f)$, where $H$ is the Shannon entropy, and $y_{A_{1:T}} = \{f(i|A_{1:t-1}) + a_t|i \in A_t, t = 1 : T\}$ denotes the collection of gain values we get from the sequence of $A_{1:T}$. We get the following regret bound:*

$$\Pr[\frac{1-e^{-1}}{k}g(S_{1:T}^*) - g(S_{1:T}) \leq \sqrt{CT\beta_T\rho_T} + C_{f,m} + 2] \geq 1 - \delta, \tag{57}$$

*Where, $C = 8/\log(1 + \sigma^{-2})$, $\beta_T = 2B^2 + 300\rho_T \ln^3(T/\delta)$.*

*Proof.* The proof directly utilizes the third case of Theorem 1 in Krause & Ong (2011), using the history sequence $(S_1, S_2, \ldots, S_t)$ as the context:

$$\Pr[\frac{1-e^{-1}}{k}g(S_{1:T}^*) - g(\hat{S}_{1:T}|S_{1:T-1}) \leq \sqrt{CT\beta_T\rho_T} + 2] \geq 1 - \delta \tag{58}$$

Combine with Lemma 1, we have:

$$\Pr[\frac{1-e^{-1}}{k}g(S_{1:T}^*) - g(S_{1:T}) \leq \sqrt{CT\beta_T\rho_T} + C_{f,m} + 2] \geq 1 - \delta. \tag{59}$$

$\square$

# B DYNAMIC INSTANCE HARDNESS (CONT.)

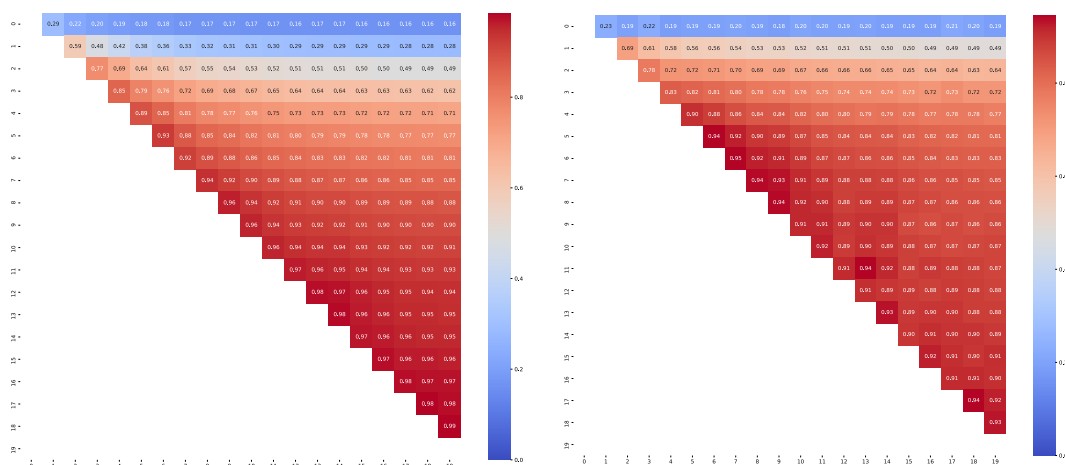

Figure 4: **LEFT:** Entry $A_{i,j}$ ($i < j$) is the percentage of shared samples between the top-$10k$ samples with the **largest** DIH computed in epoch $15i$ and epoch $15j$ **RIGHT:** Entry $A_{i,j}$ ($i < j$) is the percentage of shared samples between the top-$10k$ samples with the **smallest** DIH computed in epoch $15i$ and epoch $15j$. It shows that both the forgettable and memorable samples in the future are predictable by using the DIH values in early epochs.

Firstly, **we present a quantitative verification of the second observation in Section 2, i.e., dynamic instance hardness in early stages can predict later dynamics**. It tries to predict the samples

with large/small DIH values in the future by only using the DIH computed on early training history. In Figure 4, we show two upper triangle matrices quantitatively verifying the above statement. They are computed based on the results of the CIFAR10 training experiment in Section 2. Take the matrix $A$ in the left plot for example, given $U_i$, the top-$10k$ samples with the **largest** DIH values computed in epoch $15i$, and $U_j$ for any $j > i$, the entry $A_{i,j} = |U_i \cap U_j|/10000$. Similarly, the matrix in the right plot measures the same overlapping percentage for the top-$10k$ samples with the **smallest** DIH values between epoch $15i$ and epoch $15j$. They show that after a few first epochs, DIH can accurately predict the forgettable and memorable samples in the future. This verifies the second statement we made in Section 2. In addition, they also show that $|U_i \cap U_j|/10000$ between consecutive epochs $15i$ and $15j$ is close to 100%, which indicates that DIH is a stable and smoothly changed metric with high consistency across training trajectory.

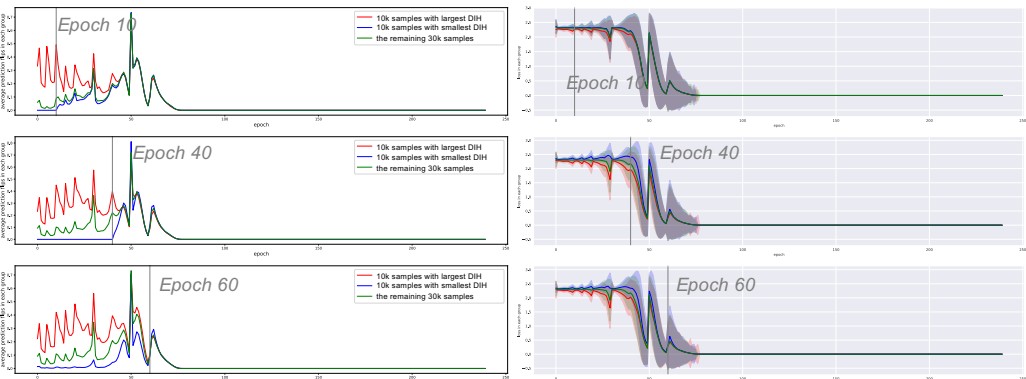

Figure 5: **LEFT:** Averaged prediction-flip and **RIGHT:** losses (mean and std.) of the three groups of samples partitioned by a DIH metric (i.e.,running mean of prediction-flip) computed at epoch 10,40 and 60 during training WideResNet-28-10 on CIFAR10 with **random labels**. In this setting, the random (but wrong) labels will be remembered very well after some training, and DIH in early stages loses the capability to predict the future DIH, i.e., they can only reflect the history but not the future. This characteristic of DIH might be helpful to detect noisy data.

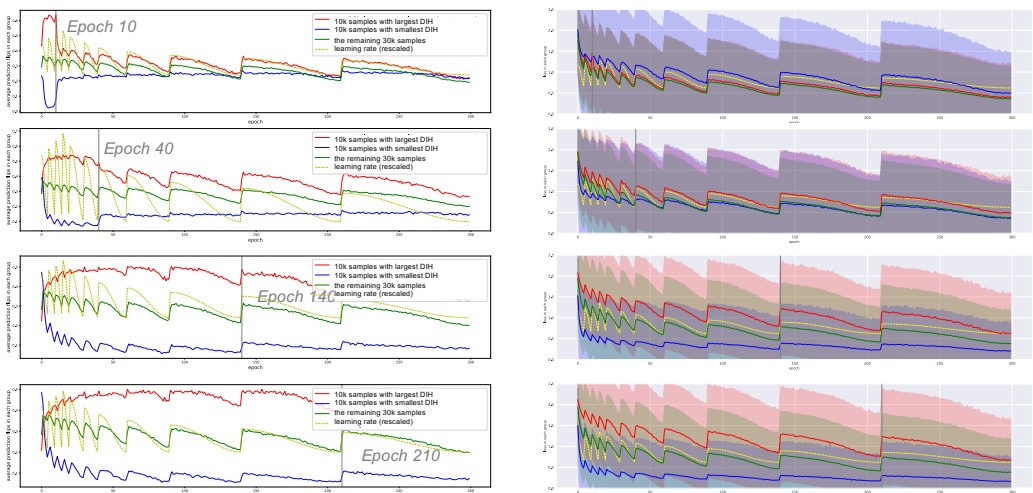

Figure 6: **LEFT:** Averaged prediction-flip and **RIGHT:** losses (mean and std.) of the three groups of samples partitioned by a DIH metric (i.e.,running mean of prediction-flip) computed at epoch 10, 40, 140 and 210 during training **a smaller CNN** on CIFAR10. It shows that the difference of memorable and forgettable samples is not sufficiently obvious until very late training epochs, e.g., after epoch-140.

Secondly, **we conduct an empirical study of dynamic instance hardness during training a neural net on very noisy data**, as studied in (Zhang et al., 2017) and (Arpit et al., 2017). In particular, we replace the ground truth labels of the training samples by random labels, and apply the same training setting used in Section 2. Then, we compute the running mean of prediction-flip for each sample at some epoch (i.e., 10, 40, 60), and partition the training samples into three groups, as we did to generate Figure 1. The result is shown in Figure 5. It shows 1) the group with the smallest

prediction flip over history (left plot) is possible to have large but unchanging loss as shown in the right plot; and 2) the DIH in this case can only reflect the history but cannot predict the future. However, it also indicates that the capability of DIH to predict the future is potential to be an effective metric to distinguish noisy data or adversarial attack from real data. We will discuss it in our future work.

Thirdly, **we change the WideResNet to a much smaller CNN architecture with three convolutional layers**[2]. We apply the same training setting used in Section 2. Then, we compute the running mean of prediction-flip for each sample at some epoch (i.e., 10, 40, 140, 210), and partition the training samples into three groups, as we did to generate Figure 1. The result is shown in Figure 6. Compared to DIH of training deeper and wider neural nets shown in Figure 1, the memorable and forgettable samples are indistinguishable until very late stages, e.g., Epoch-140. This indicates that using DIH in earlier stage to select forgettable samples into curriculum might not be reliable when training small neural nets. We will leave explanation of this phenomenon to our future works.

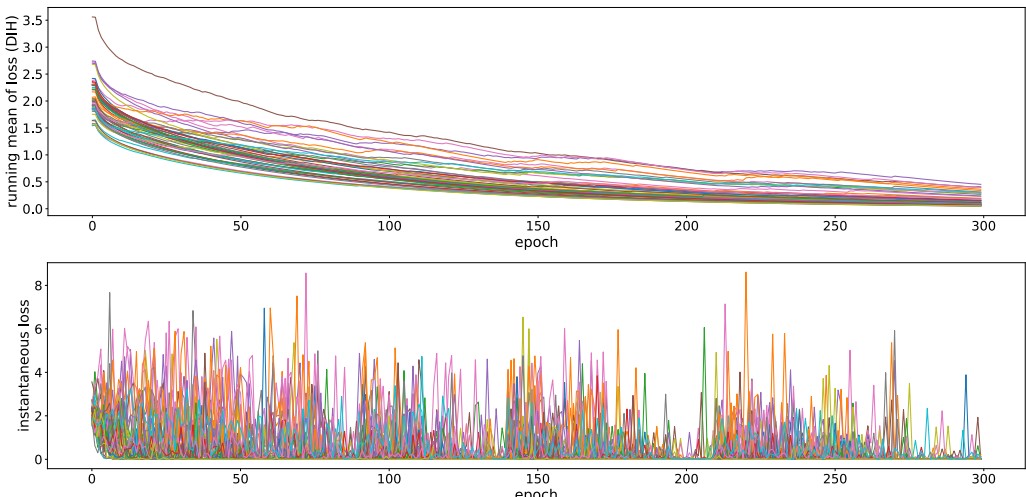

Figure 7: **Top:** DIH (running mean of loss) vs. **Bottom:** instantaneous loss of 50 randomly selected samples from CIFAR10 when training WideResNet-28-10. It shows that for each individual sample, DIH smoothly decreases while the corresponding instantaneous loss is much noisier.

Moreover, we provide a comparison of the smoothness between DIH and instantaneous loss on individual samples in Figure 7. It shows that the DIH is a smooth and consistent measure of the learning/memorization progress on individual samples. In contrast, the frequently used instantaneous loss is much noisier, so selecting training samples according to it will lead to unstable behaviors during training. In Figure 8, we also provide a comparison of DIH and instantaneous loss on the two groups of samples in Figure 2, which shows a similar phenomenon.

## C  EXPERIMENTS (CONT.)

We use cosine annealing learning rate schedule for multiple episodes. The switching epoch between each two consecutive episode for different datasets are listed below.

- CIFAR10, CIFAR100: $(5, 10, 15, 20, 30, 40, 60, 90, 140, 210, 300)$;
- Food-101, Birdsnap, FGVCaircraft, StanfordCars: $(10, 20, 30, 40, 60, 90, 150, 240, 400)$;
- ImageNet: $(5, 10, 15, 20, 30, 45, 75, 120, 200)$;
- STL10: $(20, 40, 60, 80, 120, 160, 240, 360, 560, 840, 1200)$;
- SVHN: $(5, 10, 15, 20, 30, 40, 60, 90, 140, 210, 300)$;
- KMNIST, FMNIST: $(5, 10, 15, 20, 30, 40, 60, 90, 140, 210, 300)$;

We report how the test accuracy changes with the number of training batches for each method, and the wall-clock time for all the 11 datasets in Figure 9-12.

---

[2]The "v3" network from `https://github.com/jseppanen/cifar_lasagne`.

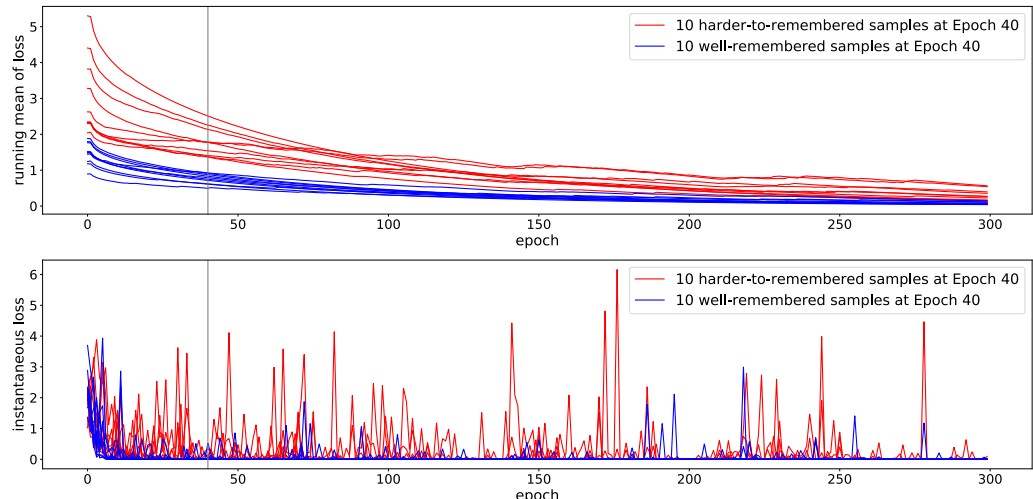

Figure 8: **Top:** DIH (running mean of loss) vs. **Bottom:** instantaneous loss of 10 samples randomly selected from the top 10k samples with the largest(red) and the smallest(blue) DIH at epoch 40 of training of WideResNet-28-10 on CIFAR10 (the same as Figure 2. It shows that for each individual sample from the two groups, DIH smoothly decreases while the corresponding instantaneous loss is much noisier.

Table 2: Details regarding the datasets and training settings (#Feature denotes the number of features after cropping if applied), "lr_start" and "lr_target" denote the starting and target learning rate for the first episode of cosine annealing schedule, they are gradually decayed over the rest episodes.

| Dataset | CIFAR10 | CIFAR100 | Food-101 | ImageNet | STL10 | SVHN |
|---|---|---|---|---|---|---|
| #Training | 50000 | 50000 | 75750 | 1281167 | 5000 | 73257 |
| #Test | 10000 | 10000 | 25250 | 50000 | 8000 | 26032 |
| #Feature | $(3, 32, 32)$ | $(3, 32, 32)$ | $(3, 224, 224)$ | $(3, 224, 224)$ | $(3, 96, 96)$ | $(3, 32, 32)$ |
| #Class | 10 | 100 | 101 | 1000 | 10 | 10 |
| #Epoch $T$ | 300 | 300 | 400 | 200 | 1200 | 300 |
| BatchSize | 128 | 128 | 80 | 256 | 128 | 128 |
| lr_start | $2 \times 10^{-1}$ | $2 \times 10^{-1}$ | $2 \times 10^{-1}$ | $2 \times 10^{-1}$ | $2 \times 10^{-1}$ | $2 \times 10^{-2}$ |
| lr_target | $5 \times 10^{-4}$ | $5 \times 10^{-4}$ | $1 \times 10^{-4}$ | $1 \times 10^{-4}$ | $5 \times 10^{-4}$ | $1 \times 10^{-3}$ |

Table 3: Details regarding the datasets and training settings (cont.)

| Dataset | Birdsnap | FGVCaircraft | StanfordCARs | KMNIST | FMNIST |
|---|---|---|---|---|---|
| #Training | 47386 | 6667 | 8144 | 50000 | 50000 |
| #Test | 2443 | 3333 | 8041 | 10000 | 10000 |
| #Feature | $(3, 224, 224)$ | $(3, 224, 224)$ | $(3, 224, 224)$ | $(1, 28, 28)$ | $(1, 28, 28)$ |
| #Class | 500 | 100 | 196 | 10 | 10 |
| #Epoch $T$ | 400 | 400 | 400 | 300 | 300 |
| BatchSize | 258 | 256 | 256 | 128 | 128 |
| lr_start | $4 \times 10^{-1}$ | $4 \times 10^{-1}$ | $4 \times 10^{-1}$ | $4 \times 10^{-2}$ | $4 \times 10^{-2}$ |
| lr_target | $1 \times 10^{-4}$ | $1 \times 10^{-4}$ | $1 \times 10^{-4}$ | $1 \times 10^{-3}$ | $1 \times 10^{-3}$ |

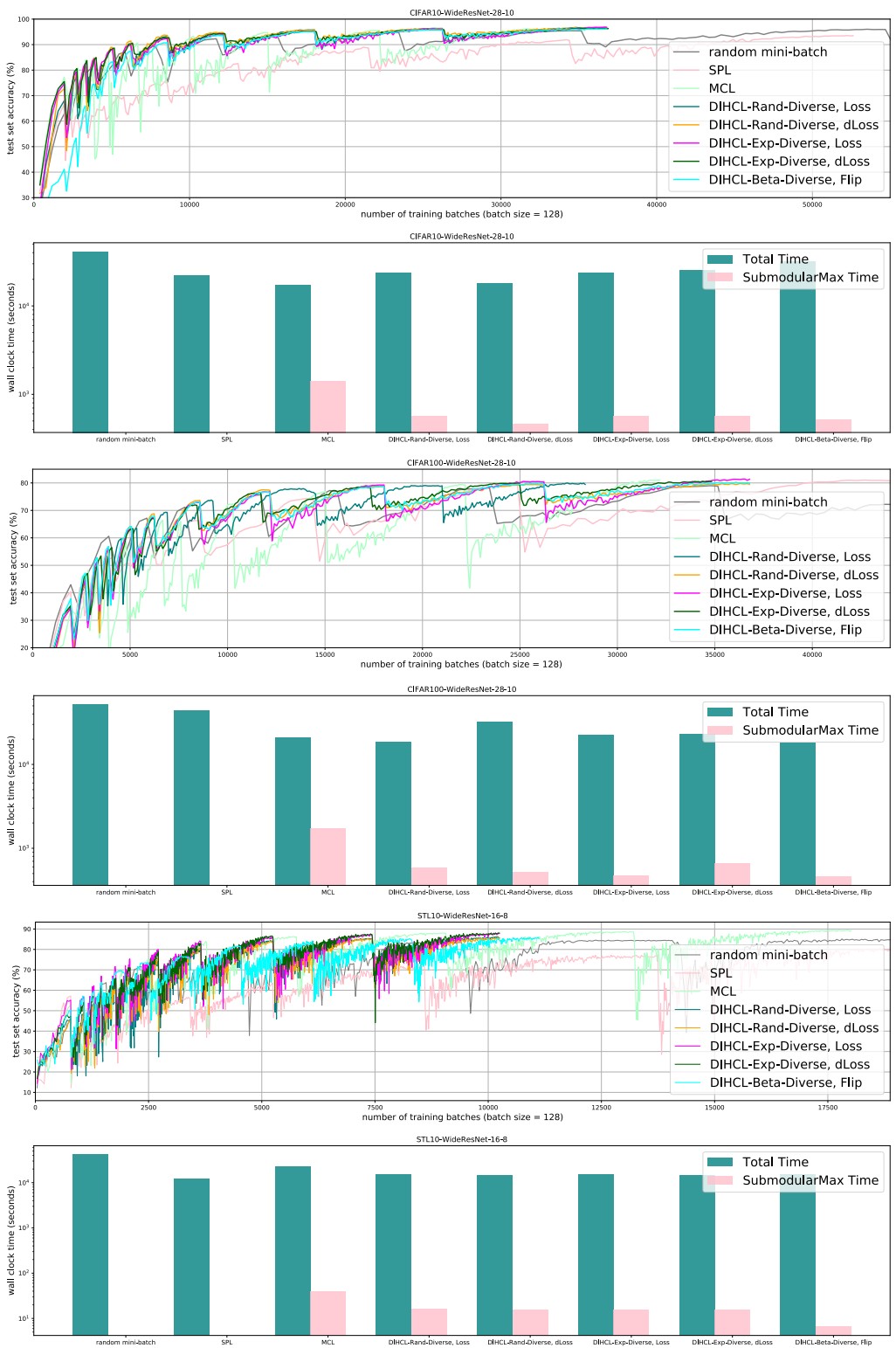

Figure 9: Training DNNs by using DIHCL (and its variants), SPL (Kumar et al., 2010), MCL (Zhou & Bilmes, 2018), and random mini-batch SGD on 3 datasets, i.e., CIFAR10, CIFAR100 and STL-10. We use "Diverse" to denote DIHCL that further reduces $S_t$ by applying submodular maximization for Eq. (5). We report how the test accuracy changes with the number of training batches for each method, and the (**log-scale**) wall-clock time for 1) the entire training and 2) the submodular maximization part in DIHCL with diversity and MCL.

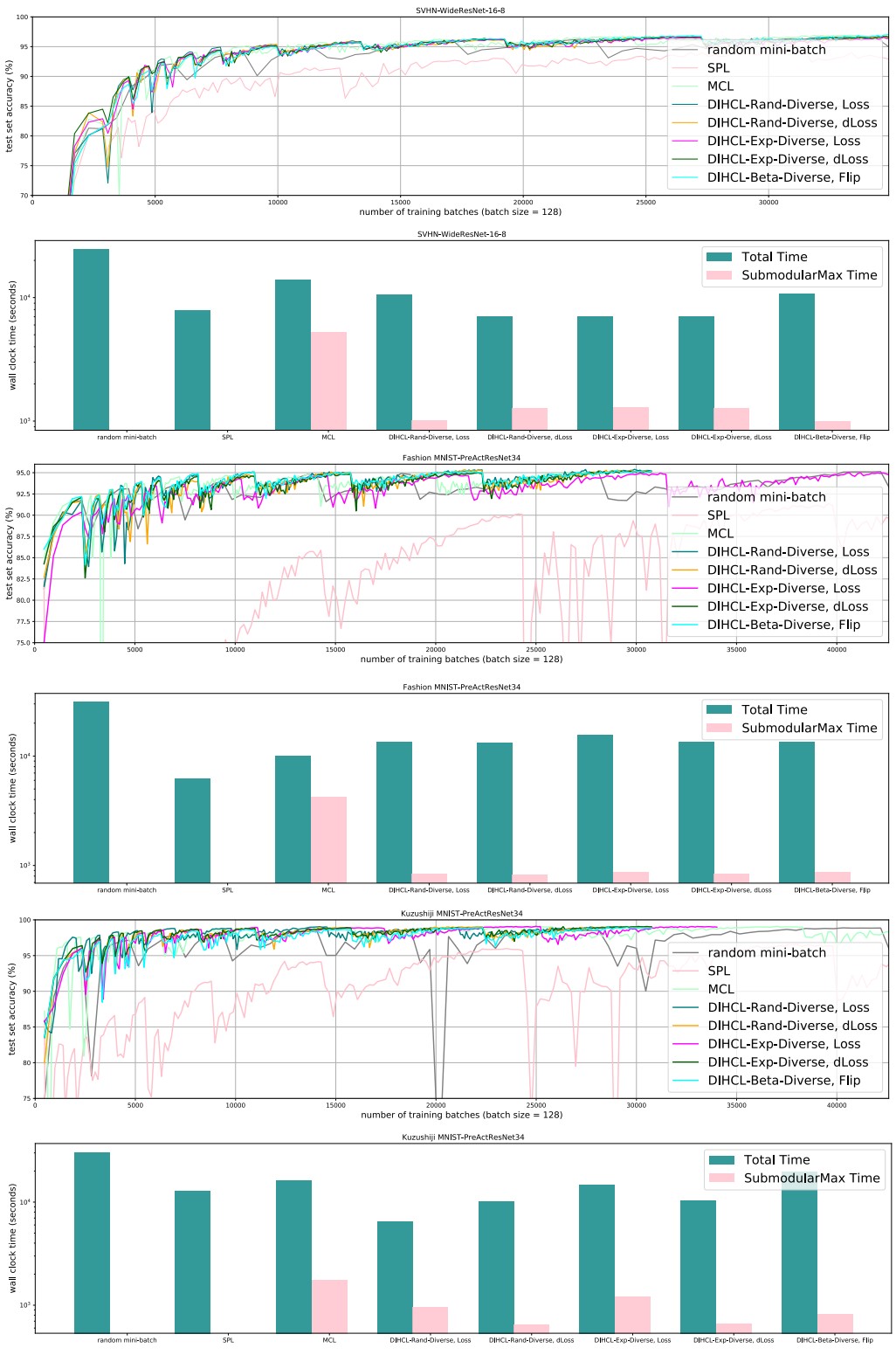

Figure 10: Training DNNs by using DIHCL (and its variants), SPL (Kumar et al., 2010), MCL (Zhou & Bilmes, 2018), and random mini-batch SGD on 3 datasets, i.e., SVHN, Fashion MNIST and Kuzushiji MNIST. We use "Diverse" to denote DIHCL that further reduces $S_t$ by applying submodular maximization for Eq. (5). We report how the test accuracy changes with the number of training batches for each method, and the (**log-scale**) wall-clock time for 1) the entire training and 2) the submodular maximization part in DIHCL with diversity and MCL.

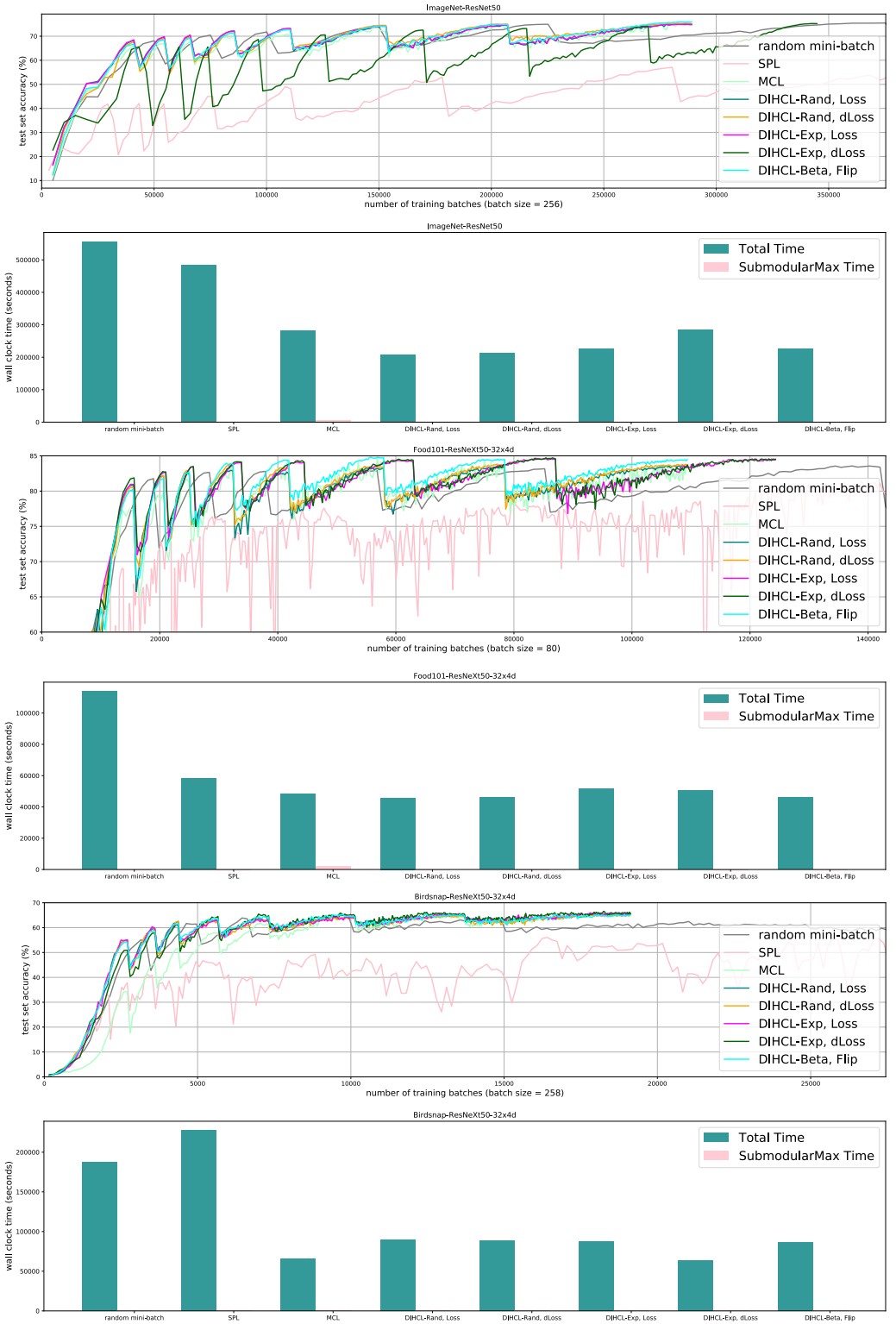

Figure 11: Training DNNs by using DIHCL (and its variants), SPL (Kumar et al., 2010), MCL (Zhou & Bilmes, 2018), and random mini-batch SGD on 3 datasets, i.e., ImageNet, Food-101 and Birdsnap. We report how the test accuracy changes with the number of training batches for each method, and the wall-clock time for 1) the entire training and 2) the submodular maximization part in MCL.

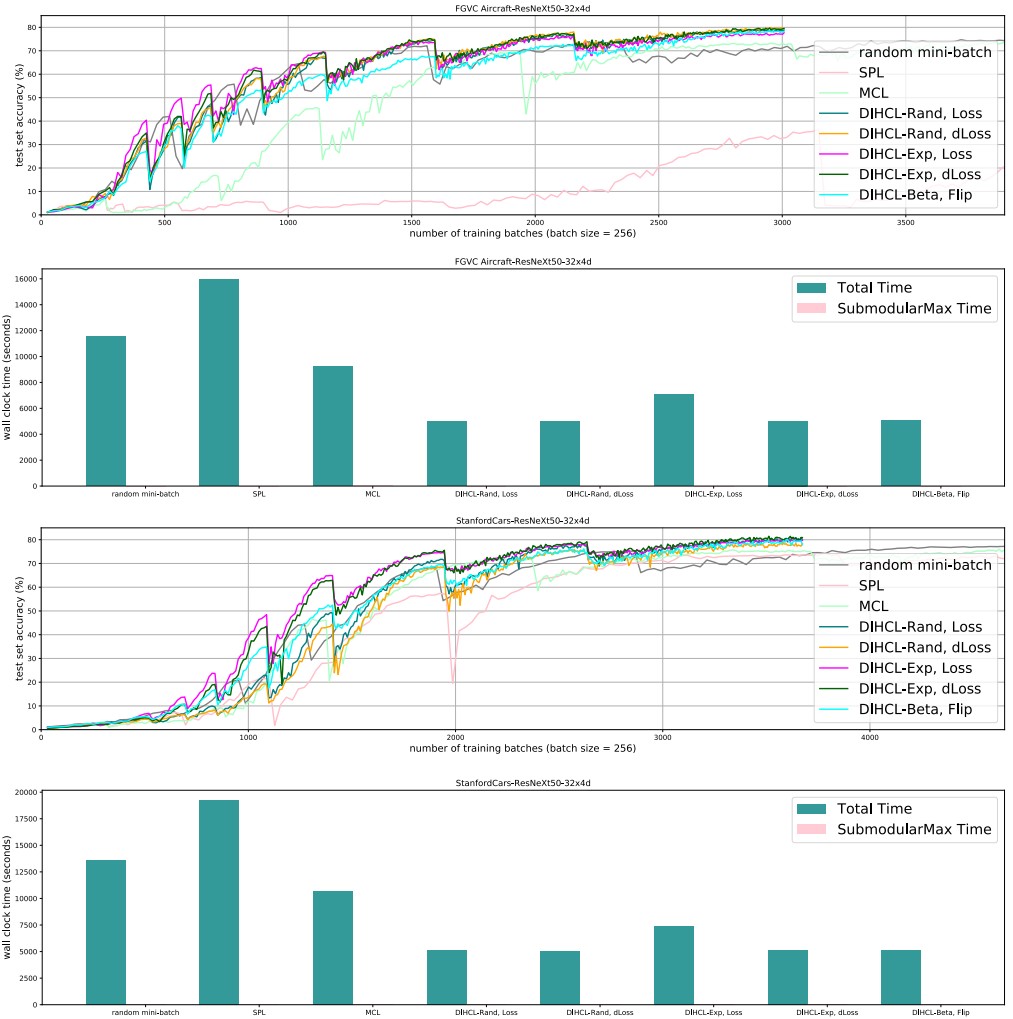

Figure 12: Training DNNs by using DIHCL (and its variants), SPL (Kumar et al., 2010), MCL (Zhou & Bilmes, 2018), and random mini-batch SGD on 2 datasets, i.e., FGVC Aircraft and Stanford Cars. We report how the test accuracy changes with the number of training batches for each method, and the wall-clock time for 1) the entire training and 2) the submodular maximization part in MCL.

