# OpenReview forum: "Dynamic Instance Hardness"
_ICLR.cc/2020/Conference — Reject_

### Official Review · AnonReviewer1 · 2019-10-22
**Official Blind Review #1**

**Rating:** 1

**Review:**

This paper studies the curriculum learning approach that can more effectively utilize the data to train DNNs. It formulates DIH as a curriculum learning problem, and derives theory on the approximation bound. The method is verified by a set of experiments. There are  several concerns raised by the reviewer.

I found the presentation of this paper is rather bad. The structure of the paper is quite strange. The Introduction section contains a lot of stuffs that I believe should be moved to the preliminary or method sections.

Also, there are a lot of confusions in the descriptions. For example, when defining the curriculum learning problem in eq.2 and eq.3, are the f's the same? If so, why do they have different input arguments?

"In step t, after the model gets trained on S_t, the feedback a_t(i) for i \in S_t is already available": I don't get this.

I am not sure what Theorem 1 tries to tell. If one chooses k large enough, the inequality satisfies trivially. BTW, what is A_{1:T}?

To sump up, there are some interesting ideas in this paper. However, with the current stage of writing, I cannot recommend acceptance.



**Experience Assessment:**

I have read many papers in this area.

**Review Assessment: Checking Correctness Of Derivations And Theory:**

I assessed the sensibility of the derivations and theory.

**Review Assessment: Checking Correctness Of Experiments:**

I assessed the sensibility of the experiments.

**Review Assessment: Thoroughness In Paper Reading:**

I read the paper at least twice and used my best judgement in assessing the paper.

---

> ### Author Response · Authors · 2019-11-15
> **Response to Review #1**
>
> Thanks for your comments! A new version of our paper has been uploaded.
>
> 1. "For example, when defining the curriculum learning problem in eq.2 and eq.3, are the f's the same? If so, why do they have different input arguments?"
>
> - It's the same f in eq.2 and eq.3. For eq.3, we are using a shorthand notation (f(i|S)) to express the gain of an element conditioning on the selected elements for function f, which is clearly defined in the 4th line on the second paragraph of page 5 (5 lines above eq.3).
>
> 2. "'In step t, after the model gets trained on $S_t$, the feedback $a_t(i)$ for $i \in S_t$ is already available': I don't get this."
>
> - After the model gets trained on $S_t$, say a forward and backward pass on a minibatch of data points when training a deep model, the losses ($a_t(i)$) on the data points in minibatch $S_t$ are a byproduct in the forward pass so we don't need to pay extra computational costs.
>
> 3. "I am not sure what Theorem 1 tries to tell. If one chooses k large enough, the inequality satisfies trivially."
>
> - Theorem 1 shows the solution of algorithm 1 is around a $1/k$ factor of the optimal solution for optimizing objective defined in eq.3. Note that $k$ is an input argument to the optimization objective defined in eq.3 and the bound in Theorem 1 holds for any input k (e.g., k=1 or k=n).
>
> In the new version of the paper, we improve the bound to the factor of $\max\{((1-e^{-1})/k, k/2n\}$. The previous bound with factor $1/k$ dominates when k is relatively small compared to n ($n > k^2$), and the $k/n$ factor dominates otherwise (when $k$ is large). We give hard examples in the first "remarks" on page 15 in the appendix showing the new factors are tight up to constant factors.
>
> 4. "BTW, what is $A_{1:T}$?"
>
> - As clearly stated in the last line of Theorem 1, $A_{1:m}$ is the argument to the "min" operator ($min_{A_{1:m}}$), which means that $A_{1:m}$ is a multiset that achieves the smallest evaluation on $C_{f,m}$.  Since the bound holds for the worst $A_{1:m}$, you can also think that the bound holds for any choice of $A_{1:m}$.

---

### Official Review · AnonReviewer2 · 2019-10-24
**Official Blind Review #2**

**Rating:** 3

**Review:**

This paper presents a technique for curriculum learning using Dynamic Instance Hardness (DIH). DIH is defined as value for each training instance that characterizes how hard that instance is and is updated throughout the training process. The DIH value is used to select the best set of instances to learn. It is shown that instances with high  (low)  DIH values maintain the high (low) value throughout the training process.

The main contributions and pros of the paper are
1. A notion of instance hardness that is persisted and updated throughout the training procedure.
2. An objective function for characterizing instance hardness as a dynamic subset selection problem. Presenting a greedy algorithm for online maximization of this function.
3. Experimental results that show how the property of instance hardness is maintained throughout training and showing how DIH-driven curriculum learning techniques that use random sampling outperforms non-curriculum learning techniques.

Cons
1. The writing of this paper is difficult and in many of the core parts of the paper, the important definitions are not clear. E..g a) the role of the function 'f' that is being maximized in section 3.2 is not clear. To elaborate, it is not obvious what this function is or why should one care about it?
2. The greedy algorithm has a bound in equation 7 that appears to be quite loose as the value of k is as high as the order of the size of the entire training set (in the experiments, 0.2n <= k < n). Am I misinterpreting it?
Furthermore, the greedy algorithm -- while being focused on the core of the technical contributions -- is not used in experimental comparisons and instead all the results presented use random sampling. At least the result of DIH-greedy should be presented.

3. This is more of a suggestion: one of the claimed advantages of the algorithm (over non curriculum learning and MCL) is that it requires less training examples to train. Given this, the authors should present training time improvements over large datasets.

**Experience Assessment:**

I have read many papers in this area.

**Review Assessment: Checking Correctness Of Derivations And Theory:**

I assessed the sensibility of the derivations and theory.

**Review Assessment: Checking Correctness Of Experiments:**

I carefully checked the experiments.

**Review Assessment: Thoroughness In Paper Reading:**

I read the paper at least twice and used my best judgement in assessing the paper.

---

> ### Author Response · Authors · 2019-11-15
> **Response to Review #2**
>
> Thanks for your comments! A new version of our paper has been uploaded.
>
> 1. "The role of the function 'f' that is being maximized in section 3.2 is not clear. To elaborate, it is not obvious what this function is or why should one care about it?"
>
> - As stated in the 5th paragraph on page 2 and formally defined in section 2.1, $f(\cdot)$ is a function that takes in a curriculum as a multiset, and returns the quality of the curriculum as a real value. Therefore, in section 3.2, we aim to find the best curriculum by maximizing $f(\cdot)$. Note that $f(\cdot)$ is unknown and inaccessible in practice as it may involve complicated dynamics of deep models and needs to produce the quality scores of all possible sequences (in the exponential number of training data). We only make assumptions about the properties of $f(\cdot)$ (the diminishing return property) and only observe the function gains as $r_t(i)$.
>
> The main purpose of formulating the problem based on $f(\cdot)$ and making these assumptions is to make a theoretical analysis of our algorithm. Under the assumptions, the function can be arbitrary and even adversarial to the curriculum selection process. As an analogy, you can think $f(\cdot)$ as the cumulation of rewards over time in the online learning setting, i.e., $f(S_1:T) = \sum_t f(S_t | S_{1:t-1})$. The reward function in online learning does not have a specific form and can even be adversarial. Under the online learning setting, we can think every data point as a bandit arm, and at every time step, we get to choose a subset of bandit arms to pull, and observe the reward of each bandit as $r_t(i) = f(i | S_{1:t})$.
>
> 2. "The greedy algorithm has a bound in equation 7 that appears to be quite loose as the value of $k$ is as high as the order of the size of the entire training set (in the experiments, $0.2n\leq k < n$). Am I misinterpreting it?"
>
> - In the new version of the paper, we improve the bound to the factor of $\max\{((1-e^{-1})/k, k/2n\}$. The previous bound with factor 1/k dominates when k is relatively small compared to $n$ (when $n > k^2$), and the $k/n$ factor dominates otherwise (when $k$ is large). We give hard examples in the first "remarks" on page 15 in the appendix showing that the new factors are tight up to constant factors. Since this is a worst-case bound, we found that the empirical performance can be much better than what the bound indicates.
>
> 3. "Furthermore, the greedy algorithm -- while being focused on the core of the technical contributions -- is not used in experimental comparisons and instead all the results presented use random sampling. At least the result of DIH-greedy should be presented."
>
> - In the first paragraph of section 3.3, we explained why randomness is essentially helpful to early exploration and accurate estimation of DIH. Note only the selected samples' DIH are updated in order to avoid extra computation. Hence, if using DIH-greedy, we will not get an accurate estimate of DIH for samples with small DIH in the first few epochs since their DIH are rarely updated.
>
> 4. "This is more of a suggestion: one of the claimed advantages of the algorithm (over non-curriculum learning and MCL) is that it requires less training examples to train. Given this, the authors should present training time improvements over large datasets."
>
> - In the appendix from page 19-22 (page 21-24 of the new version), we reported the wall clock time for DIHCL and all the baselines. It shows that DIHCL is much more efficient than other methods.

---

### Official Review · AnonReviewer3 · 2019-10-25
**Official Blind Review #3**

**Rating:** 3

**Review:**

*Revision after author response*

I thank the authors for the comments on my questions.

Unfortunately, I do not feel that these comments addressed my main concerns. For all my experimental analysis questions, the authors promised some analyses for future versions, but I was hoping to see at least a minor preliminary analysis at this point, to see if indeed my concerns are valid or not.

Moreover, for my question number 1 about the optimization problem, the authors referred me to Corollary 1 from the paper, but that didn't really help me because, as the other reviewers also point out, the writing is quite hard to follow.

Because of all these, I have decided to revise my score to a weak reject. While I believe the paper has merit, it requires revisions at many points in order for a reader to truly understand the method and trust the experimental results.

--------------------------------------------------------------------------------------------------------------
The paper proposes a curriculum learning approach that relies on a new metric, the dynamic instance hardness (DIH). DIH is used to measure the difficulty of each sample while training (in an online fashion), and to decide which samples to train on next. The authors provide extensive experiments on 11 datasets as well as some theoretical motivation for the use of this approach.

---- Overall opinion ----
Overall I believe this paper is an interesting take on curriculum learning that is able to achieve good results. I believe this approach is a combination of core ideas from multiple sources, such as boosting, self-paced learning, continual learning and other curriculum learning approaches, but overall it seems different enough from each one of them individually. Because of the resemblance with these many different methods, the method itself does not surprise through the novelty of a new idea, but the authors seemed to have found something that was missing from these methods and that leads to very good results. The experimental results look great, but I believe the paper is missing some ablation studies to assess the importance of certain components (see details below). I also had some trouble understanding certain arguments, which I hope the authors can clarify.

---- Major issues ----
1. I find the arguments section 2.1 quite difficult to follow. In particular, under the assumption stated in the paper that r_t(i) = f(i|S_{1:t−1}) =  f(e_i + S_{1:t−1}) − f(S_{1:t−1}) , why does it follow that r_t(i) can be used instead of f in the minimization problem (2).

2. Based on the method itself, it seems to me that the parameter k_t could would have a lot of influence on how well the method doing.  The authors mention in the experimental section what values they use, but there is no indication on how one would choose this value. Moreover, it would be good to see an analysis of how sensitive the results are to this choice.

3. In Figure 1, it is not clear whether the figure on the right shows the actual loss, or the smooth loss using Equation (1) with instantaneous instance (A). If it is the former, then if the loss is so smooth, why do we need DIH? If it is the latter, then what does the instantaneous loss look like? This actually raises the question of how important the smoothing component is -- could we achieve the same results with an instantaneous loss (i.e. set gamma to 1 in Eq. 1)?

---- Minor issues ----
1. How do you choose T0, gamma and gamma_k?

2. In the conclusions, the authors state that “ The reason [why  MCL and SPL are less stable] is that, compared to the methods that use DIH, both MCL and SPL deploy instantaneous instance hardness (i.e., current loss) as the score to select sample”. Since there are so many other differences in the way training progresses, I think we don’t have enough evidence to attribute this to merely the “instantaneousness” of the loss. In fact, it would be interesting to see how SPL does if you use DIH as a metric (just smoothing the loss over time), but their approach of scheduling samples (easy to hard, and not the opposite and in DIHCL).

3. Appendix C shows some interesting results regarding wall time comparison. I was surprised to see that, despite the extra computations, DHCL is comparable to random mini-batches. This makes me wonder what the stop criteria was, because when you stop matters a lot for run time comparisons. It would also be interesting to see a more ample discussion on this in the main text.

4. In Figure 1, the axes are barely readable.

5. The authors oftentimes reverse the use of \citet and \citep, for example “has been called the “instance hardness” Smith et al. (2014) corresponding to” should have a bracket, whereas “Our paper is also related to (Zhang et al., 2017)” should not have brackets.

6. This is not an issue, but I just wanted to say I appreciated Appendix B.

---- Suggestions ----
1. It would be interesting to make a connection between the DIH and what other papers have discovered about example forgetting (e.g. Toneva et. al, that was mentioned in the paper).

2. Major issues 3 -> a study on the effect of k and how to choose it.

3. While I understand that the models chosen in the experiments are expensive to train, it would be good to report standard deviations in Table 1.

4. Based on Table 1 and Figure 3, there is no concrete winner among the DIHCL methods. It would be good to include some recommendations in your conclusion on which one to choose and when.

---- Questions ----
1. “On average, the dynamics on the hard samples is more consistent with the learning rate schedule, which implies that doing well on these samples can only be achieved at a shared sharp local minima.” -> can you please explain why this is so?

2. See Major issues 3.

3. In Table 1, on some datasets, the authors apply lazier-than-lazy-greedy, and on some not.Why, and how does one decide this for a new dataset?

4. How did you choose T0, gamma and gamma_k, as well as the schedules in Appendix C (page 17)?

**Experience Assessment:**

I have published one or two papers in this area.

**Review Assessment: Checking Correctness Of Derivations And Theory:**

I did not assess the derivations or theory.

**Review Assessment: Checking Correctness Of Experiments:**

I carefully checked the experiments.

**Review Assessment: Thoroughness In Paper Reading:**

I read the paper thoroughly.

---

> ### Author Response · Authors · 2019-11-15
> **Response to Review #3**
>
> Thanks for your comments! A new version of our paper has been uploaded.
>
> 1. "I find the arguments section 2.1 quite difficult to follow. In particular, under the assumption stated in the paper that $r_t(i)=f(i|S_{1:t−1})=f(e_i+S_{1:t−1})−f(S_{1:t−1})$, why does it follow that $r_t(i)$ can be used instead of $f(\cdot)$ in the minimization problem (2)?"
>
> - We cannot directly optimize $f(\cdot)$: it is an unknown function since it measures the quality of all the possible training sequences (in exponential number) so it is intractable to be estimated. Under this assumption, $r_t(i)$ is the only observation about $f(\cdot)$, so our online optimization of $f(\cdot)$ can be only based on $r_t(i)$. We proved in Corollary 1 that only using $r_t(i)$ to optimize $f(\cdot)$ can achieve an approximation bound to the global optimum of the intractable optimization in Eq.2.
>
> 2. "It would be good to see an analysis of how sensitive the results are to the choice of $k_t$."
>
> - In our experiments over the 11 datasets, we use the same scheduling of $k_t$, i.e., we start from $k_0=n$ and exponentially reduce it to $k_t=0.2n$ by a factor $\gamma_k=0.85$, and it works well on all datasets. We will add sensitivity analysis of the scheduling parameters in future version.
>
> 3. "In Figure 1, it is not clear whether the figure on the right shows the actual loss, or the smooth loss using Eq.1 with instantaneous instance (A). If it is the former, then if the loss is so smooth, why do we need DIH? If it is the latter, then what does the instantaneous loss look like? This actually raises the question of how important the smoothing component is -- could we achieve the same results with an instantaneous loss (i.e. set gamma to 1 in Eq.1)?"
>
> - It shows the actual loss, i.e., the former case. However, each curve is the average loss over a group of samples, which makes it look smooth: for each group $V_j$ in Figure 1, Figure 1 shows how $\frac{1}{|V_j|}\sum_{i\in V_j}a_t(i)$ changes over time $t$. In Figure 7-8 of the updated version, we visualize the actual loss $a_i(t)$ and $r_i(t)$ for individual samples, which show that $a_i(t)$ is much less smooth than $r_i(t)$ on individual samples shown in Figure 2. Moreover, if we instead use $a_i(t)$ for the partition at epoch 40 as in Figure 1, we cannot see the difference between the groups in future epochs.
>
> 4. "How do you choose $T_0$, $\gamma$, $\gamma_k$, and schedules in Appendix C (page 17)?"
>
> - We tried several choices of the three parameters in $T_0\in\{5,10\}$ and $\gamma, \gamma_k\in\{0.85, 0.9, 0.95\}$ on validation sets and report the one with good performance on all the validation sets ($T_0=5, \gamma=0.95, \gamma_k=0.85$). We use the same hyperparameters for all the 11 datasets. The schedules are set based on our experience and have not been tuned.
>
> 5. "It would be interesting to see how SPL does if you use DIH as a metric (just smoothing the loss over time), but their approach of scheduling samples (easy to hard, and not the opposite and in DIHCL). It would be good to report standard deviations in Table 1."
>
> - We will report these results in our future version.
>
> 6. "I was surprised to see that, despite the extra computations, DIHCL is comparable to random mini-batches."
>
> - The training set of DIHCL in each epoch is a small subset of the one used in random mini-batches. Moreover, the extra computation of DIHCL is very cheap and only requires sorting ($O(n\log n)$) and basic arithmetic operations ($O(n)$) on one array.
>
> 7. "In Figure 1, the axes are barely readable. The authors oftentimes reverse the use of \citet and \citep."
>
> Thanks for catching this! We addressed these issues in the new version.
>
> 8. "It would be interesting to make a connection between the DIH and what other papers have discovered about example forgetting (e.g. Toneva et. al, that was mentioned in the paper)."
>
> - We discussed Toneva's work in the 4th paragraph of page 2 and its difference to ours at the end of Section 1.1. We will add more discussion.
>
> 10. "On average, the dynamics on the hard samples is more consistent with the learning rate schedule, which implies that doing well on these samples can only be achieved at a shared sharp local minima.” -> can you please explain why this is so?"
>
> - If it is a flat minima, the losses will not change when the model parameters slightly deviate from the minima (blue curves in Figure 1), so the losses will not change linearly with the learning rate, which however is the case for the hard samples (red curves) in Figure 1.
>
> 11. "In Table 1, on some datasets, the authors apply lazier-than-lazy-greedy, and on some not. Why, and how does one decide this for a new dataset?"
>
> - Lazier-than-lazy-greedy (LTLG) is applied when we need to further reduce the selected training set in Line 7 of Alg.1 by Eq.5. For the datasets we did not use LTLG, DIHCL already outperforms other baselines on both final accuracy and efficiency, so we did not apply the further reduction.

---

### Decision · Program_Chairs · 2019-12-19

**Decision:**

Reject

**Comment:**

All three reviewers, even after the rebuttal, agreed that the paper did not meet with bar for acceptance. A common complaint was lack of clarity being a major problem. Unfortunately, the paper cannot be accepted in its current form. The authors are encouraged to improve the presentation of their approach  and resubmit to a new venue.